# A cleavable chimeric peptide with targeting and killing domains enhances LPS neutralization and antibacterial properties against multi-drug resistant *E. coli*

Zhenlong Wang[1,2], Da Teng[1,2], Ruoyu Mao[1,2], Ya Hao[1,2], Na Yang[1,2], Xiumin Wang [1,2,3✉] & Jianhua Wang [1,2,3✉]

Pathogenic *Escherichia coli* is one of the most common causes of diarrhea diseases and its characteristic component of the outer membrane-lipopolysaccharide (LPS) is a major inducer of sepsis. Few drugs have been proven to kill bacteria and simultaneously neutralize LPS toxicity. Here, the chimeric peptides-R7, A7 and G7 were generated by connecting LBP14 (LPS-targeting domain) with L7 (killing domain) via different linkers to improve antibacterial and anti-inflammatory activities. Compared to parent LBP14-RKRR and L7, the antibacterial activity of R7 with a cleavable "RKRR" linker and the "LBP14-RKRR + L7" cocktail against *Escherichia coli*, *Salmonella typhimurium* and *Staphylococcus aureus* was increased by 2~4-fold. Both A7 and G7 with non-cleavable linkers almost lost antibacterial activity. The ability of R7 to neutralize LPS was markedly higher than that of LBP14-RKRR and L7. In vivo, R7 could be cleaved by furin in a time-dependent manner, and release L7 and LBP14-RKRR in serum. In vivo, R7 can enhance mouse survival more effectively than L7 and alleviate lung injuries by selective inhibition of the NF-κB signaling pathways and promoting higher IAP activity. It suggests that R7 may be promising dual-function candidates as antibacterial and anti-endotoxin agents.

[1] Team of AMP & Alternatives to Antibiotics, Gene Engineering Laboratory, Institute of Feed Research, Chinese Academy of Agricultural Sciences, Beijing 100081, People's Republic of China. [2] Key Laboratory of Feed Biotechnology, Ministry of Agriculture and Rural Affairs, Beijing 100081, People's Republic of China. [3] These authors jointly supervised this work: Xiumin Wang, Jianhua Wang. ✉email: wangxiumin@caas.cn; wangjianhua@caas.cn

Pathogenic *Escherichia coli*, a Gram-negative bacterium, causes ~1.7 billion cases of diarrhea in both humans and animals worldwide each year[1]. Worryingly, multidrug-resistant (MDR) *E. coli* strains were difficult to be killed by the last-resort antibiotics, polymyxins[2,3]. Furthermore, pathogenic Gram-negative bacterial infections are always accompanied by the release of lipopolysaccharide (LPS) from the cell wall[4,5]. LPS, also known as endotoxin, is a prominent component of the outer membranes of Gram-negative bacteria and construct a prime natural barrier that restricts the entry of antibacterial agents into bacteria[6,7]. Additionally, LPS serves as a crucial virulence factor known as pathogen-associated molecular pattern (PAMP)[8,9], which typically stimulates immune cells to activate related signaling pathways, triggering the activation of related signaling pathways and causing an uncontrollable cytokine storm. These factors contribute to the development of severe sepsis with a mortality rate of 30 ~ 50%[10–12]. Commonly used antibiotics, such as ampicillin and polymyxin B, cannot effectively neutralize LPS[13,14] due to lack of specific binding sites and LPS modifications, and even accelerate LPS release from bacteria[15,16]. Therefore, there is an urgent need to design a novel candidate capable of killing MDR *E. coli*, neutralizing LPS toxicity, and inhibiting the LPS-induced downstream cascade.

In recent years, it has been paid considerable attention to antimicrobial peptides (AMPs) owing to their potent activity, low cytotoxicity and low tendency to develop bacterial resistance[17,18]. Among them, chimeric peptides are comprised of two different dominant peptides, and they have been proven to be an effective strategy to improve the activity and specificity of AMPs. Tan et al. designed two chimeric peptides, 13 and 15, by linking a phage-displayed peptide with a narrow-spectrum peptide using the "GGG" linker or no linker. Compared to the parental peptides, chimeric peptide 13 with the "GGG" linker exhibited a 5.0- or 2.4-fold increase in activity or specificity. Although chimeric peptides 15 without a linker retained similar activity against *E. coli*, its specificity was lower than that of peptide 13[19]. In our previous work, we designed chimeric peptides-A6 and G6 by connecting AMP-N6 with LPS-targeting peptide LBP14 via either a rigid linker ("(EA$_3$K)$_2$") or a flexible linker ("G$_4$S"). Both of them specifically killed *E. coli* and simultaneously neutralized LPS toxicity due to the chimera of LBP14 and L7; however, LBP14-L7 without a linker only exhibited very weak activity[13]. Therefore, the presence of an appropriate linker is crucial for the biological activity of the chimeric peptides. The currently developed linkers are classified into non-cleavable and cleavable ones in vivo[20]. Non-cleavable linkers between functional domains can provide some advantages such as prolonging plasma half-life, improving expression, and targeting specific sites in vivo. For example, the stability and catalytic activity of the fusion protein Glu-Xyl were significantly improved after the insertion of the non-cleavable linkers ("(EA$_3$K)n")[21]. Compared to non-cleavable linkers, the cleavable linkers have advantages in reducing steric hindrance, increasing bioactivity, and allowing the resulting two parts to perform independent functions after cleavage[20]. In previous studies, some cleavable linkers have been successfully applied in cleavable chimeric peptides or proteins, including IFN-RKRR-HAS, immunotoxin, FIX-albumin, and P14KanS, respectively, which could be cleaved by enzymes or in reducing environments to improve the efficacy in vivo[22–25].

Bovine lactoferricin (LfcinB) is a cationic AMP consisting of 25 amino acid residues derived from the N-terminus of bovine lactoferrin, It has a broad-spectrum antibacterial effect on both Gram-negative and Gram-positive bacteria and shows minimal cytotoxicity[26,27]. L7 (17 residues), a derivative of LfcinB developed in our laboratory, exhibits good antibacterial activity against Gram-negative and Gram-positive bacteria, particularly *E. coli*,

but its LPS-neutralizing activity is weak[28]. The LBP14 peptide, a fragment of residues 86-99 of the LPS binding protein (LBP), has a strong ability to bind to LPS[13,29]. In this study, to enhance their biological activity, cleavable and non-cleavable chimeric peptides-R7, A7 and G7 were designed by connecting LBP14 (targeting domain) with L7 (killing domain) via the cleavable ("RKRR") and non-cleavable ("(EA$_3$K)$_2$"" or "G$_4$S") linkers, respectively. We hypothesized that cleavable chimeric peptide R7 can be cleaved by furin, a protease mainly localized in the cell surface, endosomes, and golgi apparatus in eukaryote[30,31]. Thus, R7 would be cleaved in vivo and release free LBP14 and L7, respectively, which may have higher biostability and biological functions. Conversely, non-cleavable chimeric peptides-A7 and G7 cannot be cleaved by furin. The antibacterial/anti-endotoxin activity, serum stability, and toxicity of these chimeric peptides were evaluated in detail, followed by their in vivo therapeutic efficacy in mice challenged with MDR *E. coli* and LPS, respectively.

## Results

### Design of cleavable/non-cleavable chimeric peptides with targeting/killing domains and their physicochemical properties.

To improve LPS-neutralizing ability of antibacterial peptide L7, cleavable chimeric R7 peptide was designed by connecting L7 with LPS-binding peptide LBP14 via cleavable linker "RKRR". Meanwhile, two non-cleaved chimeric peptides-A7 and G7 were designed by connecting LBP14 with L7 via either a rigid linker "(EA$_3$K)$_2$" or a flexible linker "G$_4$S"[13], which were served as control to compare the effect of cleavable and non-cleavable linkers on the activity of chimeric peptides. The amino acid sequences and characteristic of cleavable chimeric peptide R7 and its parental peptides are summarized in Table 1. Theoretically, cleavable chimeric peptide R7 may be cleaved by furin protease in vivo at the last R (arginine) of the "RKRR" linker, resulting in the release free LBP14-RKRR and L7, while non-cleavable chimeric peptides-A7 and G7 keep intact.

The GRAVY of all three chimeric peptides (-0.615 ~ −1.103) was lower than that of parent-L7 (-0.594), and AI values of R7 (42.00) and G7 (40.83) except A7 (50.49) was also lower than that of L7 (46.47). Additionally, R7 (+14), A7 (+10), and G7 (+10) have more positive charges than parents (+5), indicating their more potent interaction with bacterial membranes by electrostatic attraction. BI provides an overall estimate of the potential of a peptide to bind to membranes[32]. BI value of R7, LBP14 and LBP14-RKRR was larger than L7, A7 and G7, indicating their stronger interaction with bacterial membranes; it may be related to LBP14 binding to LPS.

### Cleavable chimeric peptide R7 exhibits more potent and broad-spectrum antimicrobial activity.

As shown in Table 2, the MIC values of cleavable chimeric peptide R7 against MDR *E. coli* CVCC195 was 1.13 μM, lower than those of LBP14 (4.53 μM), LBP14-RKRR (2.26 μM), and L7 (2.26 μM), but equal to their combinations ("LBP14 + L7", 1.13 μM; "LBP14-RKRR + L7", 1.13 μM). The MIC of R7 against *S. typhimurium* CVCC533 (1.13 μM) is significantly lower than those of parental peptides or combinations (4.53-18.1 μM). It is worth noting that the "LBP14-RKRR + L7" and "LBP14 + L7" combinations exhibited the same activity against MDR *E. coli* CVCC195 and *S. typhimurium* CVCC533. Moreover, the activity of R7 and the "LBP14-RKRR + L7" combination against the tested *S. aureus* strains was increased by ≥2-fold compared to parent peptides (LBP14-RKRR and L7). It indicates that cleavable chimeric peptide R7 and the "LBP14-RKRR + L7" combination displayed more potent antimicrobial activities against Gram-negative and Gram-positive bacteria than their parent peptides (LBP14-RKRR and L7). In

**Table 1 Sequence and physicochemical properties of cleavable/non-cleavable chimeric peptides.**

| Peptide | Sequence | Characteristic | MW (Da) | Charge (+) | pI | GRAVY | AI | BI (kcal mol⁻¹) | Hydrophobicity |
|---|---|---|---|---|---|---|---|---|---|
| LBP14 | RVQGRWKVRASFFK | LPS-binding peptide | 1765.10 | 5 | 12.31 | -0.793 | 48.57 | 3.26 | 0.236 |
| LBP14-RKRR | RVQGRWKVRASFFK-RKRR | LPS-binding peptide with cleavable linker | 2361.83 | 9 | 12.70 | -1.583 | 37.78 | 5.33 | -0.039 |
| L7 | FKAWRWAWRMKKLAAPS | Antibacterial peptide | 2133.59 | 5 | 12.02 | -0.594 | 46.47 | 1.49 | 0.494 |
| R7 | LBP14-RKRR-L7 | Chimera of LBP14 and L7 via a cleavable linker (RKRR) | 4477.41 | 14 | 12.85 | -1.103 | 42.00 | 3.46 | 0.220 |
| A7 | LBP14-$(EA_3K)_2$-L7 | Chimera of LBP14 and L7 via a rigid linker ($(EA_3K)_2$) | 4821.72 | 10 | 11.86 | -0.615 | 50.49 | 2.07 | 0.251 |
| G7 | LBP14-$G_4S$-L7 | Chimera of LBP14 and L7 via a flexible linker ($G_4S$) | 4195.96 | 10 | 12.61 | -0.656 | 40.83 | 1.96 | 0.324 |

"RKRR": cleavable linker, "$(EA_3K)_2$" and "$G_4S$": non-cleavable linker, MW molecular weight, pI isoelectric point, GRAVY grand average of hydropathicity, AI aliphatic index, BI Boman index.

**Table 2 MIC and MBC values of the peptides.**

| Strains | LBP14 | | LBP14-RKRR | | L7 | | R7 | | LBP14 + L7 [a] | | LBP14-RKRR + L7 [a] | | PMB | | A7 [b] | | G7 [c] | |
|---|---|---|---|---|---|---|---|---|---|---|---|---|---|---|---|---|---|---|
| | MIC (μM) | MBC (μM) | MIC (μM) | MBC (μM) | MIC (μM) | MBC (μM) | MIC (μM) | MBC (μM) | MIC (μM) | MBC (μM) | MIC (μM) | MBC (μM) | MIC (μM) | MBC (μM) | MIC (μM) | MBC (μM) | MIC (μM) | MBC (μM) |
| Gram-negative bacteria | | | | | | | | | | | | | | | | | | |
| Escherichia coli CVCC195 [d] | 4.53 | 4.53 | 2.26 | 2.26 | 2.26 | 4.53 | 1.13 | 1.13 | 1.13 | 2.26 | 1.13 | 2.26 | 1 | 1 | 36.2 | 36.2 | 18.1 | 18.1 |
| S. typhimurium CVCC533 | 18.1 | 18.1 | 4.53 | 128 | 18.1 | 18.1 | 1.13 | 2.26 | 4.53 | 4.53 | 4.53 | 4.53 | 8 | 8 | >36.2 | >36.2 | >36.2 | >128 |
| Gram-positive bacteria | | | | | | | | | | | | | | | | | | |
| Staphylococcus aureus ATCC43300 [b] | 36.2 | 36.2 | 4.53 | 64 | 18.1 | 18.1 | 2.26 | 2.26 | 9.06 | 9.06 | 2.26 | 2.26 | >64 | >64 | >36.2 | >36.2 | >36.2 | >36.2 |
| S. aureus CVCC546 [b] | 36.2 | >36.2 | 9.06 | 9.06 | 18.1 | 36.2 | 2.26 | 2.26 | 4.53 | 9.06 | 2.26 | 2.26 | >64 | >64 | >36.2 | >36.2 | >36.2 | >36.2 |

[a] 1:1 mixture of LBP14 (or LBP14-RKRR) and L7.
[b] A7: connect LBP14 and L7 via rigid linker $(EA_3K)_2$ [13].
[c] G7: connect LBP14 and L7 via flexible linker $G_4S$ [13].
[d] The strains are resistant to different antibiotics [13,62]. The MDR E. coli CVCC195 strain is resistant to doxycycline, benzocillin, penicillin, lincomycin, clindamycin, vancomycin, chloramphenicol, erythromycin, amikacin, streptomycin, gentamicin and sulfamethoxazole, respectively. The S. aureus ATCC43300 strain is resistant to lincomycin, amoxicillin, amikacin, ampicillin, oxacillin, erythrocin, sulfisoxazole, neomycin, azithromycin, kanamycin, gentamicin, and penicillin, respectively. The S. aureus CVCC546 strain is resistant to tetracycline, bacitracin, and sulfisoxazole, respectively.

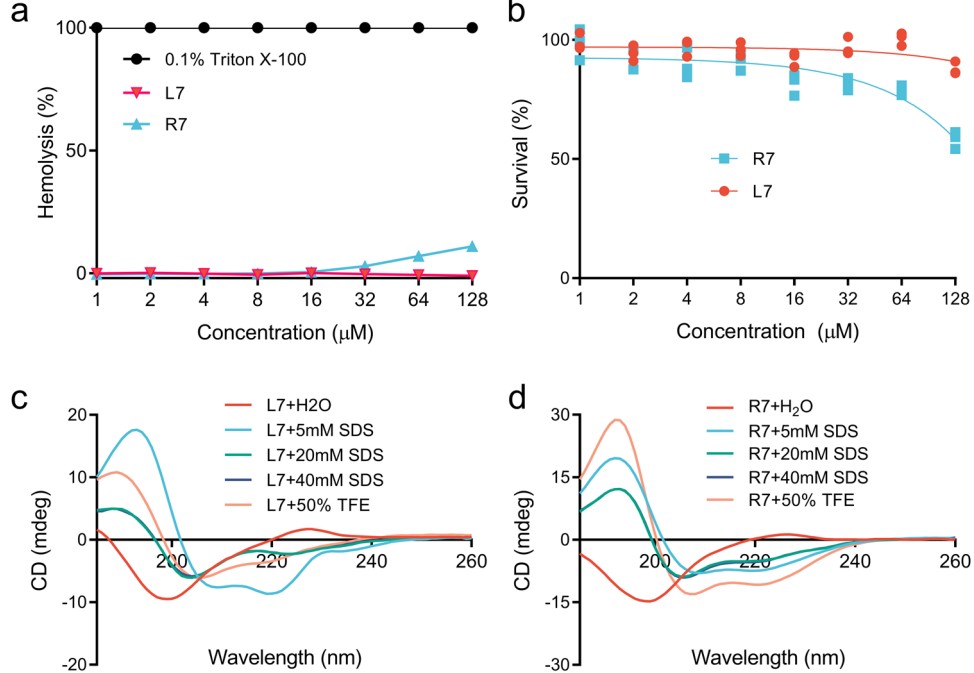

**Fig. 1 The hemolysis, toxicity and CD spectra of cleavable chimeric peptide R7 and its parent L7. a** Hemolytic activity of R7 and L7 against mouse erythrocyte. **b** The cytotoxic activity of R7 and L7 against RAW 264.7 monocytes by MTT assy. CD spectra of L7 (**c**) and R7 (**d**) in water, SDS, and 50% TFE, respectively. Results indicate means with SD ($n = 3$ independent experiments).

contrast, non-cleavable chimeric peptides-A7 and G7 exhibited low antibacterial activity (≥18.1 μM). It suggests that the cleavable linker ("RKRR") may be more suitable for the chimerization of L7 with LBP14 than the non-cleavable linkers, including "$(EA_3K)_2$" and "$G_4S$".

**Cleavable chimeric peptide R7 has moderate hemolysis and cytotoxicity.** As shown in Fig. 1a, similarly to its parent L7, cleavable chimeric peptide R7 has no hemolysis and cytotoxicity at low concentration (0 ~ 32 μM). However, at a higher concentration of 128 μM, the hemolysis of R7 and L7 against mouse red blood cells was 10.9% and 0%, respectively. Additionally, The RAW 264.7 cell survival of L7 and R7 at the concentration of 128 μM was 87.7% and 58.2%, respectively. These results suggested that R7 has higher toxicity than L7 at high concentrations (Fig. 1b).

**Secondary structures of cleavable chimeric peptide R7 and its parent L7.** TFE is usually used to simulate the hydrophobic environment of eukaryotic cells, while SDS is often used to simulate the environment of bacterial cell membranes. To investigate the structural features of L7 and R7, the CD spectra of peptides were measured in $ddH_2O$, 5 ~ 40 mM SDS and 50% TFE buffer, respectively. The secondary structures of L7 and R7 in $ddH_2O$ were characterized predominantly by antiparallel, β-turn and random coils with a characteristic positive maximum at 230 nm and a negative minimum at 200 nm (Fig. 1c, d). In SDS and 50% TFE buffer, chimeric peptide R7 exhibited a significant increase in α-helical content (22.9 ~ 50.4%) with a characteristic positive peak at 190 nm and two negative peaks at 208 and 222 nm, respectively, higher than the parent peptide L7 (10.1 ~ 32.0%). Particularly, α-helix of R7 in 50% TFE was up to 50.4% higher than L7 (13.9%) (Supplementary Table 1 and Supplementary Table 2). These results suggest that R7 is more prone to form α-helix conformation in cell membrane environment.

**Intact L7 is released from cleavable chimeric peptide R7 in vitro.** To investigate whether chimeric peptide R7 can be cleaved by furin in vitro, R7 was incubated with furin for 65 h. The result showed that R7 could be cleaved correctly by furin in a time-dependent manner (Fig. 2a and Supplementary Fig. 1), and the intact LBP14-RKRR and L7 fragments were also released from R7. The cleavage efficiency of R7 was just 36.23% (24 h) and 61.84% (65 h), respectively. It indicates that intact L7 peptide may be successfully released from cleavable chimeric peptide R7 after the cleavage.

The release of L7 from chimeric peptide R7 was further confirmed by MALDI-TOF MS. As shown in Fig. 2b, two separate product peaks were found and they could match theoretical peptide fragments of L7 (with the corresponding MW = 2133.174 Da) and LBP14-RKRR (MW = 2361.423 Da). These data suggest that the intact LBP14-RKRR and L7 peptide fragments can be successfully released from cleavable chimeric peptide R7 after the furin cleavage, and they may exhibit their independent biological activities and functions.

**Cleavable chimeric peptide R7 improves serum stability and releases L7 in serum.** The degradation of R7 was evaluated by RP-HPLC after incubation in mouse or human serum. As shown in Fig. 3a, b, after incubation for 2 ~ 8 h in mouse serum, 29.8 ~ 21.7% of R7 remained intact, which was higher than that of L7 (25.3 ~ 0.8%). Meanwhile, the antibacterial activity of L7 against MDR E. coli CVCC195 declined significantly after incubation for 4 ~ 8 h in mouse serum; and completely lost at 8 h (Fig. 3c). In contrast, antibacterial activity of R7 did not changed in mouse serum until 8 h. These results suggest that R7'activaty is more stable in mouse serum than L7.

To determine whether R7 can be properly cleaved in mouse serum, MALDI-TOF MS was used to detect the released products from of R7 after incubation in mouse serum. It was found a signal of L7 product peak, but the product peak signal of

**a**

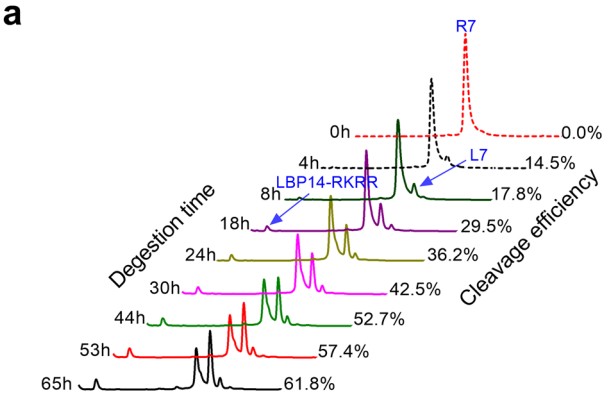

**b**

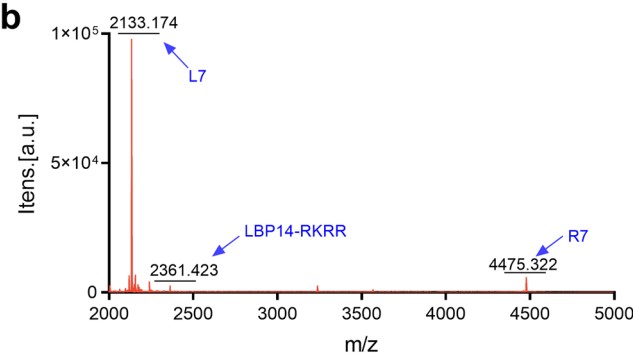

**Fig. 2 Release of L7 from cleavable chimeric peptide R7 in vitro. a** The cleavage rate of R7 after the incubation with furin. The result was repeated twice and one typical result was shown. **b** The MALDI-TOF MS of R7 incubated with furin.

released-LBP14-RKRR was not detected (Fig. 3d), which may be due to the weak ionized ability of LBP14-RKRR[33]. Furthermore, we used more sensitive HPLC-MS method to analyze the peptide mapping of R7 after incubation in mouse serum, and found that L7 were present at 45.43 min (Supplementary Table 3). Similar results were found in human serum (Supplementary Fig. 2). These results indicate that R7 can release active L7 and LBP14-RKRR in mouse or human serum.

**Cleavable chimeric peptide R7 binds to LPS.** In the BC displacement assay, the BC displacement rate always reflect the ability of peptides to interact with LPS[34]. As shown in Fig. 4a, all peptides (including R7, L7, LBP14, LBP14-RKRR and peptide combinations) induced an increase in fluorescence in a concentration-dependent manner. Notably, R7 had a significantly stronger affinity with LPS than other peptides. The BC displacement rate of chimeric peptide R7 was 96.05%, higher than that of the combinations of "LBP14-RKRR + L7" (88.39%) and "LBP14 + L7" (80.64%); they were superior to single parental peptides-L7 (71.33%), LBP14-RKRR (70.05%), and LBP14 (55.71%), respectively. This result suggests that chimeric peptide R7 and its parental peptide combinations (1:1) have a stronger ability to bind to LPS than L7, LBP14-RKRR, and LBP14 alone. It is worth noting that the "LBP14-RKRR + L7" combination has greater ability to bind to LPS than the "LBP14 + L7" combination in the case of 1:1 mixture.

A putative binding site between peptides and LPS was predicted by molecular docking (MD). As shown in Fig. 4b, five hydrogen bonds were formed between chimeric peptide R7 and LPS at Lys7–DAO1012, Ser11–FTT1011, Arg15–PO42001, Arg15–MYR1014, and Arg18–GMH1005 while L7 only formed

two hydrogen bonds with LPS at Lys11–PO42001 and Ala15–KDO1002. These results suggest that R7 may have stronger affinity with LPS compared to its parent L7. This is mainly attributed to the attachment of the LPS-targeting domain LBP14-RKRR, which can bind to LPS by the formation of four hydrogen bonds (Arg1–KDO1002, Gly4–PO42001, Ser11–DAO1012 and Arg15–DPO2000). Significantly, the binding complex conformation of chimeric peptide R7 and LPS showed that LBP14-RKRR (the head of R7) can bind to LPS, but L7 (the tail of R7) may have no binding site, which is in according with our design expectations that the introduction of LPS-targeting LBP14 in chimeric peptides can increase the ability of R7 to bind to LPS.

**Cleavable chimeric peptide R7 protects mice from MDR _E. coli_ challenge.** Our previously constructed mouse intraperitoneal infection model was used to investigate the in vivo antimicrobial activity of peptides[13]. We first used R7 as a therapeutic agent against _E. coli_ and LPS infected mouse, and the results showed that R7 was almost ineffective (≤20%) in treatment groups (Supplementary Fig. 3). Therefore, we changed our thinking and applied R7 to prevention experiments. The mice were intraperitoneally injected with 7 μmol/kg (minimum effective dose in preliminary test (Supplementary Fig. 4)) chimeric peptide-R7, L7, LBP14-RKRR, the "LBP14-RKRR + L7" combination or PBS, followed by an intraperitoneal injection with a lethal dose of MDR _E. coli_ CVCC195 at 6 h. The mice only injected with MDR _E. coli_ died within 2 days; however, the mouse survival in the R7-preventive treatment group was 100% at 7 days, higher than that of L7 (60%), "LBP14-RKRR + L7" (80%), and LBP14-RKRR (0%), respectively (Fig. 5a). It indicates that R7 has higher antibacterial activity in mice than the parental peptides and their combination.

The viable bacterial counts from infected lungs after treatment with peptides were shown in Fig. 5b. Compared to the _E. coli_ group without any treatment, all peptides significantly reduced the bacterial counts in the lungs. At 12 h, the bacterial counts in the _E. coli_ group were 4.97 $\log_{10}$CFU/g; preventive treatment with 7 μmol/kg R7 and "LBP14-RKRR + L7" reduced the bacterial counts by 2.71 and 2.75 $\log_{10}$CFU/g, respectively. Treatment with 7 μmol/kg parent peptides only resulted in a decrease of 0.92 (LBP14-RKRR) and 0.57 $\log_{10}$CFU/g (L7). Similar results were also found at 24 h, R7 and "LBP14-RKRR + L7" reduced the bacterial counts by 2.82 and 2.84 $\log_{10}$CFU/g, higher than LBP14-RKRR (1.46 $\log_{10}$CFU/g) and L7 (1.47 $\log_{10}$CFU/g). This indicates that preventive treatment with chimeric peptide R7 or the "LBP14-RKRR + L7" combination was more effective at killing bacteria than parent peptides, such as LBP14-RKRR or L7 alone.

Bacterial LPS is the main cause of inflammatory response[35]. To analyze the effects of the peptides on cytokines, TNF-α, IL-6, IL-1β and IL-10 levels in mouse serum were detected after _E. coli_ challenge at 12 h or 24 h. As shown in Fig. 5c–f, when the mice were only challenged with _E. coli_ (_E. coli_ group), the levels of proinflammatory cytokines of TNF-α, IL-6, and IL-1β, which involved in the upregulation of inflammatory responses, increased to 93.525 ~ 96.423, 149.549 ~ 158.702, and 41.607 ~ 46.486 pg/mL, respectively at 12 ~ 24 h. Compared to _E. coli_ group, after preventive treatment with R7 and "LBP14-RKRR + L7", the levels of TNF-α, IL-6 and IL-1β were significantly decreased by 41.1 ~ 46.1%, 22.2 ~ 35.0%, and 41.6 ~ 49.1%, respectively at 12 h; higher than those of parent peptide LBP14-RKRR (14.7%, 14.3%, and 15.5%) and L7 (21.7%, 17.4%, and 34.6%) (Fig. 5c–e). In _E. coli_ group, the level of anti-inflammatory cytokine IL-10 reduced to 13.492 ~ 15.499 pg/mL.

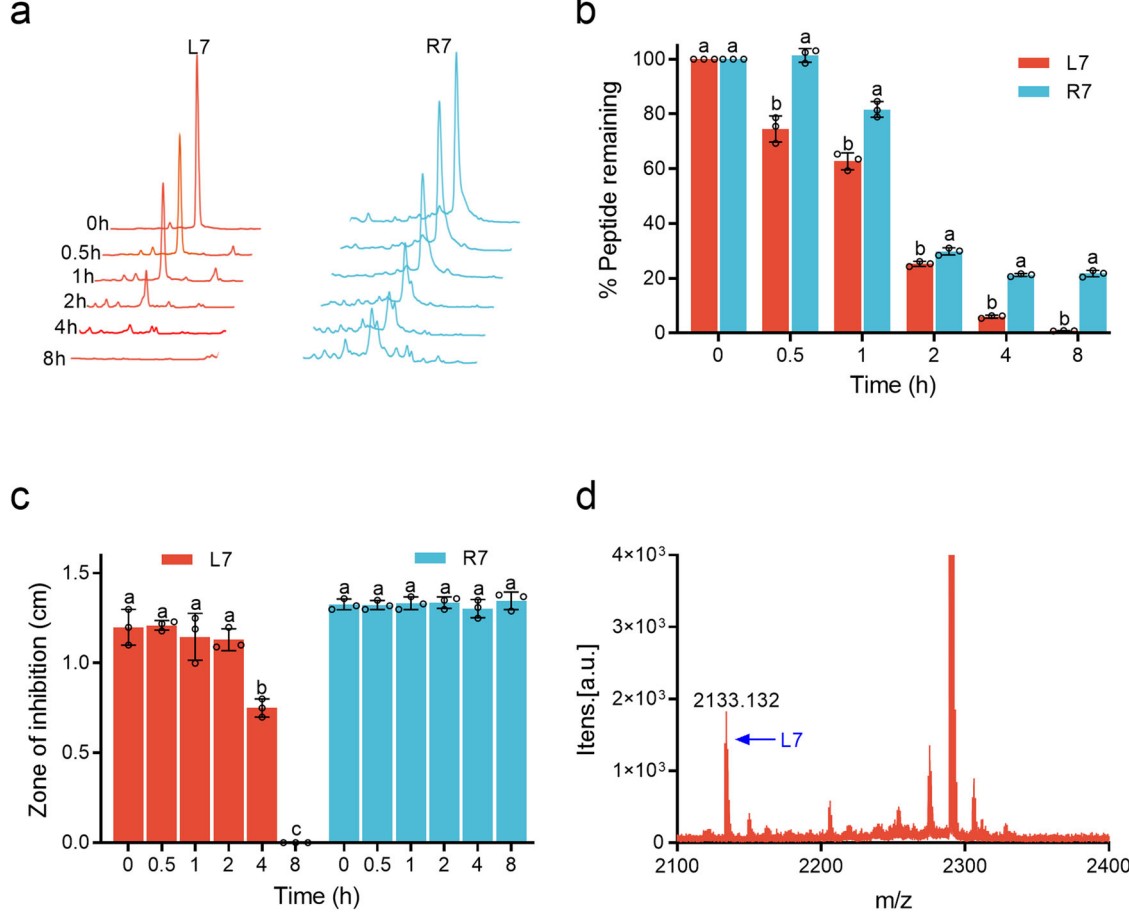

**Fig. 3 Stability and release of peptides in mouse serum. a** Detection of the peak area change of chimeric peptide R7 and its parent L7 in serum by RP-HPLC. **b** The remaining rate of R7 and L7 in mouse serum. **c** The antibacterial activity of R7 and L7 against MDR *E. coli* in mouse serum detected by an inhibition assay. **d** The MALDI-TOF MS of R7 after incubation in mouse serum for 2 h. The experiments were repeated three times. Data in (**a–d**) is representative of three biological replicates. The results are given as the mean ± SD of three independent experiments in (**b**, **c**). Different lower-case letters indicate a significant difference between the two groups ($p < 0.05$).

After treatment with R7 and "LBP14-RKRR + L7", the IL-10 level was significantly increased by 37.4 ~ 60.8%, higher than those of parent peptides such as LBP14-RKRR (11.4%) and L7 (21.0%) (Fig. 5f). Additionally, after treatment with R7 and "LBP14-RKRR + L7", the levels of TNF-α, IL-6, and IL-1β had decreased by 42.0 ~ 45.7%, 27.3 ~ 36.4%, and 42.7 ~ 47.1%, respectively at 24 h, higher than those of LBP14-RKRR (14.2%, 12.7%, and 12.3%) and L7 (28.5%, 17.8%, and 34.9%) (Fig. 5c–e); the IL-10 levels in R7 and "LBP14-RKRR + L7" treatment group were increased by 52.4 ~ 76.1%, higher than those of LBP14-RKRR (25.7%) and L7 (36.4%) (Fig. 5f). These data suggest that cleavable chimeric peptide R7 and the "LBP14-RKRR + L7" combination can more potently regulate immune system than parent peptides (LBP14-RKRR and L7) by down-regulating pro-inflammatory factors and up-regulating anti-inflammatory factors.

Intestinal alkaline phosphatase (IAP) can relieve LPS toxicity by dephosphorylate LPS[16]. To investigate the effect of peptides on IAP levels, the duodenum of experimental mice was collected and IAP activity was detected by ELISA. As shown in Fig. 5g, after infection with *E. coli* for 12 h, the IAP activity of mice in *E. coli* group reduced to 1006.0 U/g. The IAP activity in mice pre-treated with peptides were 457.5 U/g (LBP14-RKRR), 646.8 U/g (L7), 836.0 U/g ("LBP14-RKRR + L7"), and 514.6 U/g (R7), respectively. These results indicate that the "LBP14-RKRR + L7" combination can better relieve LPS toxicity than R7, L7, and

LBP14-RKRR by increasing IAP expression. There was no markedly change in *E. coli* infection group and preventive treatment groups at 24 h.

Whether L7 or R7 protects mice from MDR *E. coli*-induced lung injury was analyzed histologically at 12 h or 24 h after infection. The certain degree of lung injury of each mouse is described by a score (asymptomatic: 0 point, mild: 1 point, moderate: 2 points, severe: 3 points). As shown in Fig. 6a, b, acute lung injury was found in *E. coli* group with 9.0 points at 12 h, characterized by thickened alveolar septum and infiltration of inflammatory cells, and was not significantly improved in LBP14-RKRR (8.0 points) and L7 (7.7 points) group. In contrast, pre-treatment with 7 μmol/kg R7 and the "LBP14-RKRR + L7" combination markedly reduced lung damage with the scores of 4.3 and 5.7 points. Lung injury in *E. coli* group further worsened with the 12.7 points at 24 h. In contrast, R7 and "LBP14-RKRR + L7" markedly alleviated lung damage with the scores of 2.0 and 3.7 points, which were more effective than LBP14-RKRR (8.0 points) and L7 (5.7 points) group.

Moreover, all peptides reduced infiltration of inflammatory cells in the lung; the inflammatory cells in per mm² were decreased by 64.2 ~ 69.0% (R7), 48.5 ~ 49.73% ("LBP14-RKRR + L7"), 18.2 ~ 30.5% (LBP14-RKRR), and 17.4 ~ 22.7% (L7) respectively. (Fig. 6c). The result indicates that chimeric peptide R7 were more effective at alleviating lung injury in *E. coli*-infected mice than L7 and the "LBP14-RKRR + L7" combination.

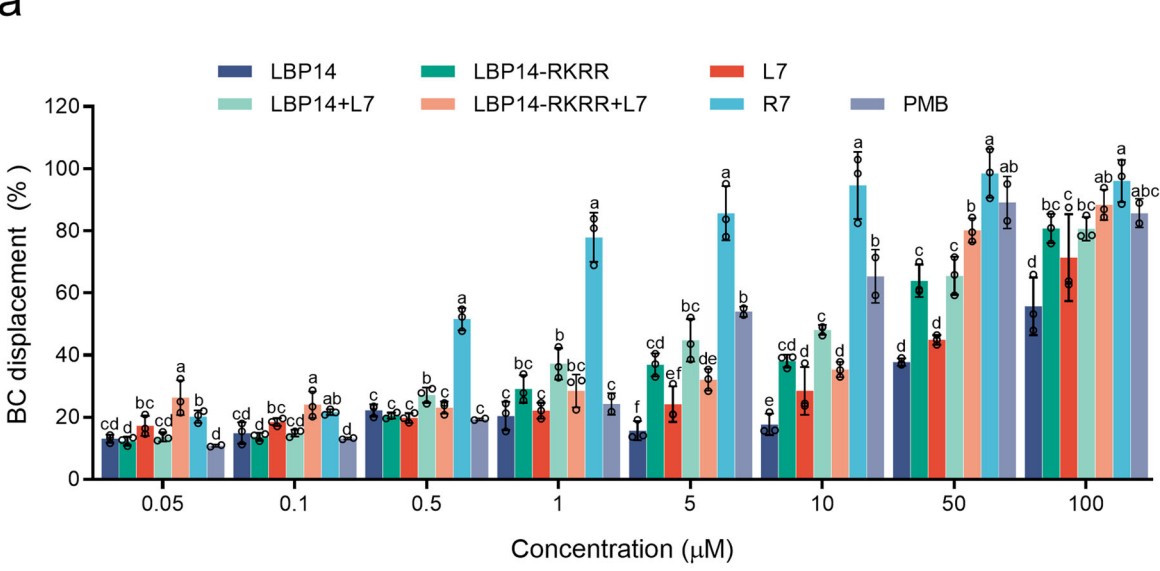

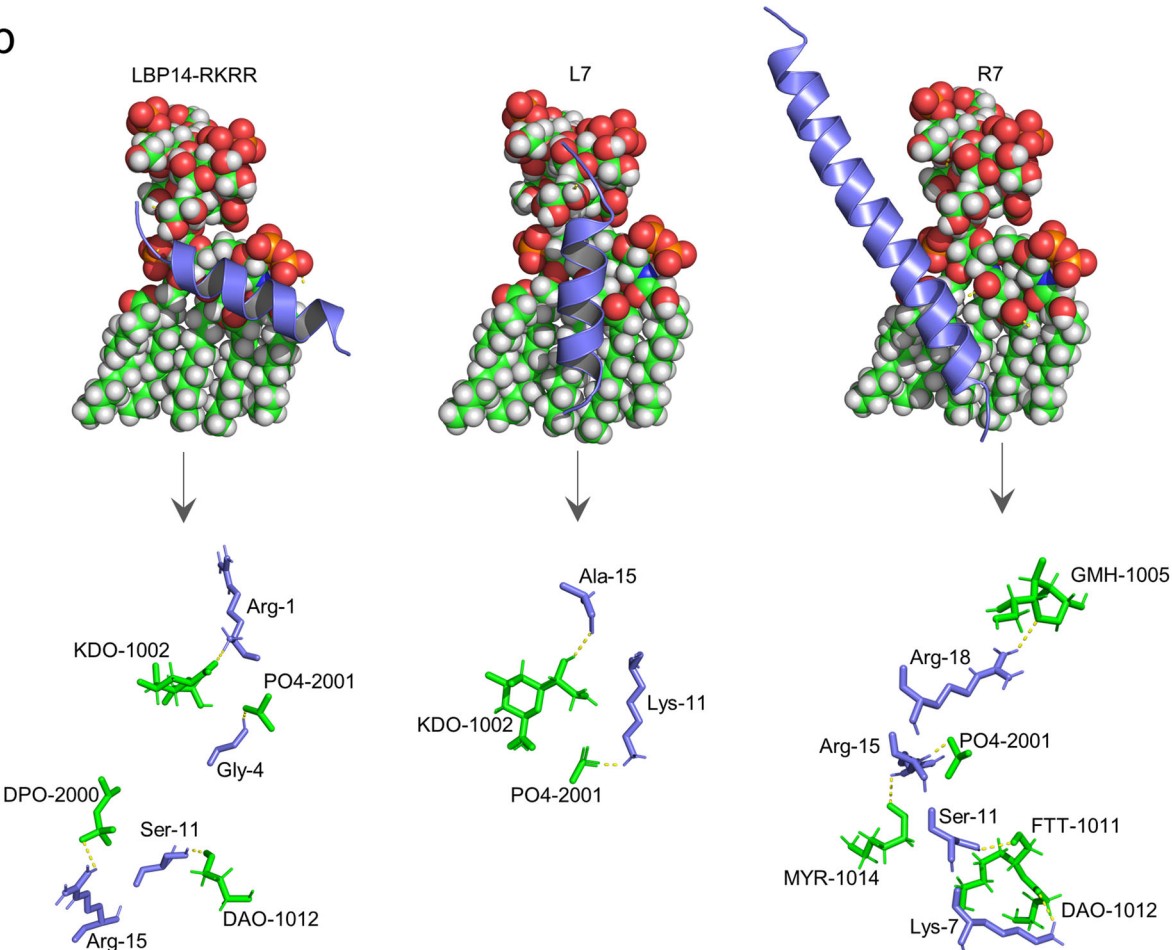

**Fig. 4 Interaction between peptides and LPS. a** Binding affinity of R7, L7, LBP14, LBP14-RKRR, "LBP14-RKRR + L7", "LBP14 + L7", and PMB to LPS. The results are given as the mean ± SD of three independent experiments. Different lower-case letters indicate a significant difference between the two groups ($p < 0.05$). **b** Putative binding sites of LBP14-RKRR, L7, and R7 with LPS. Docking was performed using Autodock vina. Up: complex structures of LBP14-RKRR, L7 or R7 and LPS. Oxygen, carbon, hydrogen atoms are indicated as red, green and white, respectively. Down: H-bonding between the peptides and LPS. LPS and peptide chains are shown as green and blue, respectively. The hydrogen bonds are shown in yellow dotted lines.

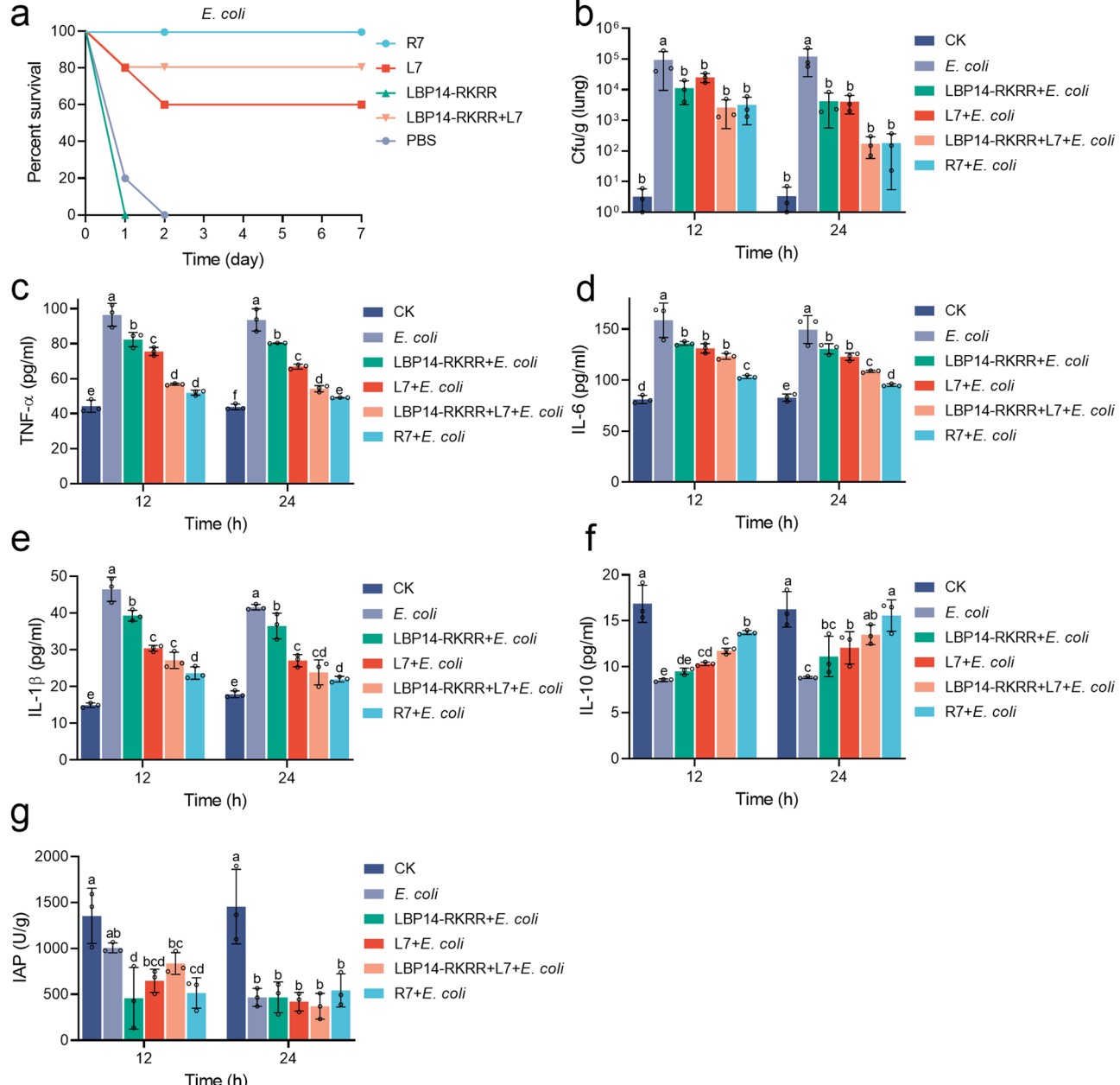

**Fig. 5 Efficacy of cleavable chimeric peptide R7 in the mice challenged with MDR *E. coli*. a** Survival of mice. Mice were intraperitoneally injected with 7 µmol/kg different peptides (including R7, L7, LBP14-RKRR, the 'LBP14-RKRR + L7' combination) at 0 h (five mice/group), followed by injection with MDR *E. coli* CVCC195 (0.5 × 10^9 CFU/mouse) at 6 h. Survival was recorded for 7 days. Two biological replicates were done. **b** Effects of peptides on the bacterial counts in lungs. Mice were infected intraperitoneally with MDR *E. coli* CVCC195 (0.5 × 10^9 CFU/mouse) and pre-treated with 7 µmol/kg different peptides at 6 h (three mice/group) before infection, respectively. Lungs were removed at 12 h and 24 h after infection to analyze bacterial translocation. **c–g** Effects of peptides on cytokines (**c–f**) and the IAP levels in mice (**g**). The results are given as the mean ± SD of three independent experiments. Different lower-case letters indicate a significant difference between the two groups (*p* < 0.05).

**Cleavable chimeric peptide R7 protects mice from lethal challenge with LPS**. It was also evaluated the efficacy of peptides in mice challenged with LPS. The mice in the negative control (LPS group) died within 2 days. In the R7-preventive group, the mouse survival was up to 100% at 7 days, which was higher than those of the "LBP14-RKRR + L7" combination (40%), L7 (40%), and LBP14-RKRR (20%), respectively (Fig. 7a). This result indicates that chimeric peptide R7 more effectively protects the mice from LPS-induced death than LBP14-RKRR, L7 and the "LBP14-RKRR + L7" combination.

A significant increase in TNF-α (71.2 ~ 82.3 pg/mL), IL-6 (137.6 ~ 141.3 pg/mL) and IL-1β (35.4 ~ 37.2 pg/mL) was observed in LPS-injected mice, as compared to a blank control group (CK group) (Fig. 7b–d). The concentrations of serum TNF-α, IL-6 and IL-1β in LPS-challenged mice pre-treated with R7 were 43.1 ~ 46.1, 92.5 ~ 99.3, and 16.9 ~ 17.5 pg/mL, respectively; levels of TNF-α, IL-6 and IL-1β in LPS-challenged mice pre-treated with "LBP14-RKRR + L7" were 51.9 ~ 53.4, 104.9 ~ 110.3, and 18.7 ~ 21.6 pg/mL, respectively, which were significantly lower than those from the corresponding

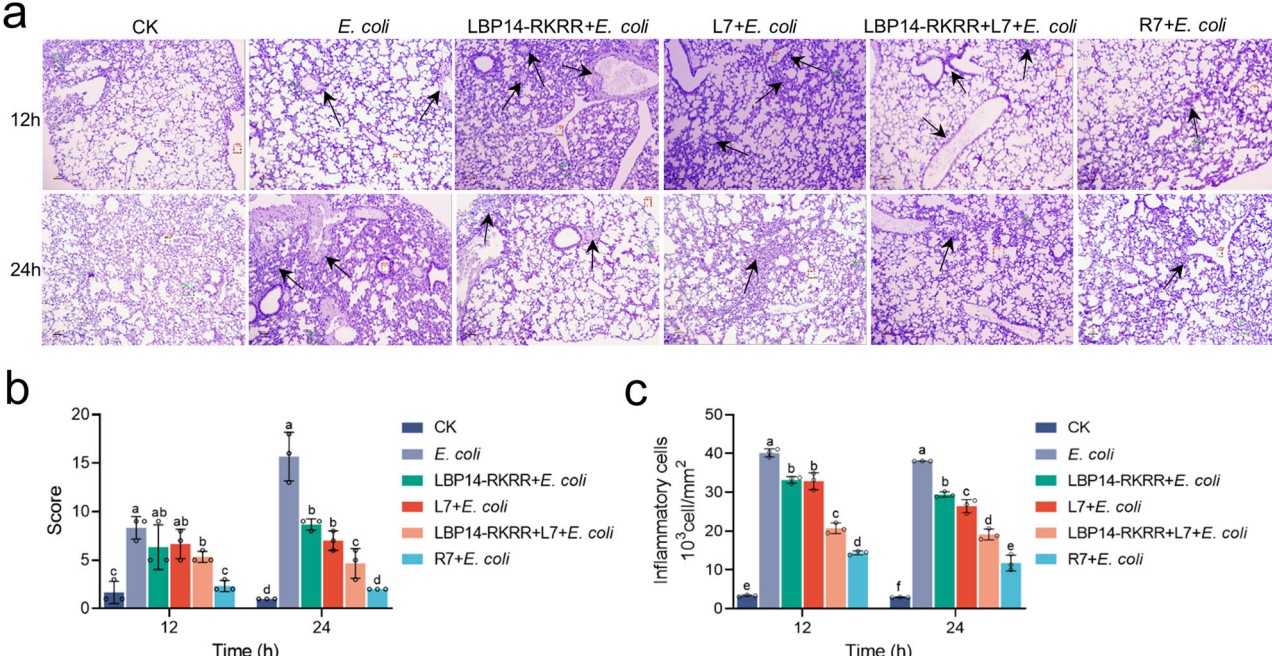

**Fig. 6 Effects of cleavable chimeric peptide R7 on lung injury in mice.** Mice were intraperitoneally injected with 7 μmol/kg different peptides (including R7, L7, LBP14-RKRR, and the "LBP14-RKRR + L7 combination, respectively) at 0 h (three mice/group), followed by injection with MDR *E. coli* CVCC195 ($0.5 \times 10^9$ CFU/mouse) at 6 h. Lung injury was analyzed histologically at 12 h or 24 h. **a** Histopathological images of lung sections after staining with hematoxylin and eosin (H&E). Images are presented at a magnification of 100×. **b** Pathological analysis of lung. Lung injury involves in inflammation (such as peribronchial, perivascular, intrabronchial, and pulmonary interstitial one), bronchial exudate, and alveolar septal thickening, respectively. The specific degree of lung injury of each mouse is assessed by Score 0 (asymptomatic), Score 1 (mild), Score 2 (moderate), and Score 3 (severe), respectively. **c** The number of counted infiltrated inflammatory cells within processed lung sections from **a**. Data represent the mean ± SD of three independent experiments. Different lower-case letters indicate a significant difference between the two groups ($p < 0.05$).

parental LBP14-RKRR group (65.5 ~ 75.9, 126.9 ~ 130.3, and 28.6 ~ 30.4 pg/mL, respectively) and L7 group (56.4 ~ 68.0, 113.2 ~ 120.2, and 24.6 ~ 26.1 pg/mL, respectively). These data suggest that pre-treatment with cleavable chimeric peptide R7 can more effectively inhibit the production of proinflammatory cytokines in LPS-challenged mice than LBP14-RKRR, L7 or the "LBP14-RKRR + L7" combination.

As shown in Fig. 7e, the concentration of serum IL-10 was 13.9 ~ 14.5 pg/mL for R7 group and was 12.2 ~ 12.8 pg/mL for "LBP14-RKRR + L7" group, higher than that from the corresponding parental LBP14-RKRR group (10.6 ~ 12.1 pg/mL) and L7 group (11.5 ~ 12.0 pg/mL). The results indicate that pre-treatment with R7 in LPS-challenged mice can more effectively promote anti-inflammatory cytokine level than LBP14-RKRR, L7 or "LBP14-RKRR + L7".

As shown in Fig. 7f, after challenge with LPS, the IAP activity in mice reduced to 541.4 ~ 595.7 U/g at 12 h or 24 h, respectively. Compared to the LPS group, R7 significantly up-regulated the activity of IAP at 12 h, superior to LBP14-RKRR, L7, and the "LBP14-RKRR + L7" combination, but no significant changes were found at 24 h. The result indicates that R7 can more potently attenuate LPS toxicity than LBP14-RKRR, L7 and the "LBP14-RKRR + L7" combination by improving the IAP activity.

Lung injury was also analyzed in LPS-challenged mice. As shown in Fig. 8a, b, in the negative control (LPS group), the mean score of lung injury was 8.0 ~ 14.0 points. Pathological scores were not significantly improved in the pre-treatment with peptide groups compared to LPS group at 12 h. In contrast, pre-treatment with 7 μmol/kg R7 and "LBP14-RKRR + L7" markedly reduced lung inflammation with the scores of 3.0 and 4.0 points, respectively, lower than those of LBP14-RKRR (8.3 points) and L7 (6.3 points). Moreover, both R7 and "LBP14-RKRR + L7"

reduced infiltration of inflammatory cells in the lung; the inflammatory cells in per mm$^2$ were decreased by 69.8% (12 h) and 69.9% (24 h) for R7, and by 48.3% (12 h) and 47.9% (24 h) for L7 (Fig. 8c), which were higher than those of LBP14-RKRR (13.3 ~ 16.5%) and L7 (33.8 ~ 41.8%). These results indicate that R7 was more effective at alleviating lung injury in LPS-challenged mice than its parent peptides (LBP14-RKRR and L7) and the LBP14-RKRR + L7" combination. Furthermore, the effects of R7 and L7 on LPS-induced signal pathways involved in inflammation were detected by western blotting. Compared to L7, R7 significantly promoted (27.2%) LPS-induced IκBα levels and inhibited (47.0%) the LPS-induced translocation of the NF-κB p-p65 subunit from the cytoplasm to the nucleus (Supplementary Fig. 5a–c and Supplementary Fig. 6). However, ERK1/2 phosphorylation was not markedly attenuated by R7 and L7 (Supplementary Fig. 5a, d and Supplementary Fig. 6), suggesting that R7 and L7 didn't regulate MAPK pathway. These results demonstrate that R7 exert the anti-inflammatory effects through the selective inhibition of the NF-κB inflammatory signaling pathways.

## Discussion

Gram-negative bacterium *E. coli* is one of the most common causes of diarrhea diseases. LPS is a major constituent of the outer leaflet of Gram-negative bacteria and can trigger life-threatening sepsis[36]. Antibiotics usage, while facilitating clearance of pathogenic bacterial infections, also can contribute to the development of bacterial antibiotic resistance and accelerates the release of LPS[15]. LPS is a target for the development of novel antibacterial agents against Gram-negative bacteria. In this study, we designed dual-functional peptides by connecting LBP14 (binding to LPS)

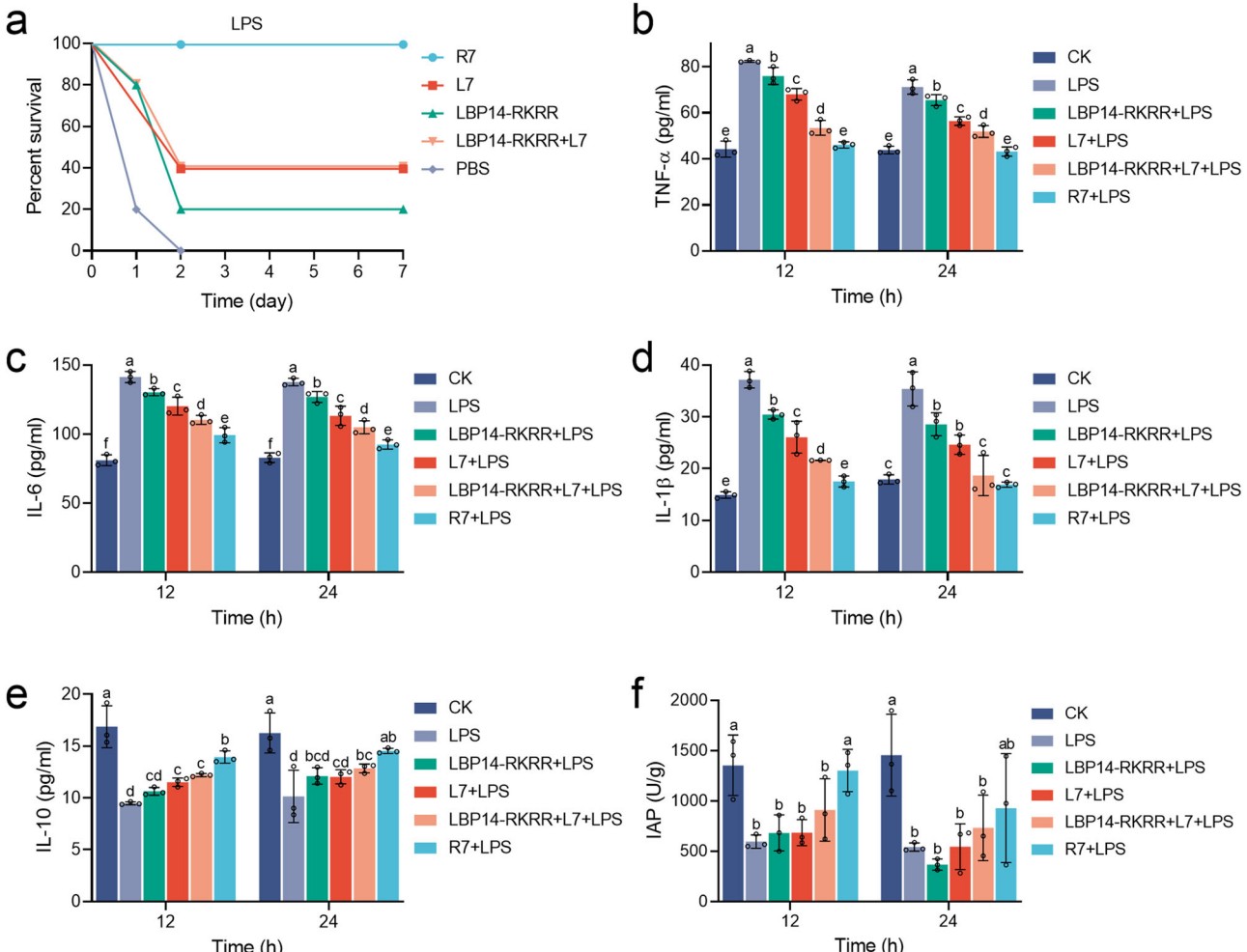

**Fig. 7 Efficacy of cleavable chimeric peptide R7 in the mice challenged wiht LPS. a** Survival of mice. Mice were intraperitoneally injected with different peptides (7 μmol/kg) at 0 h (five mice/group), followed by injection with LPS (13.5 mg/kg of body weight) at 6 h. Survival was recorded for 7 days. Two biological replicates were done. **b–f** Effects of L7 and R7 on cytokines (**b–e**) and the IAP levels (**f**). The results are given as the mean ± SD of three independent experiments. Different lower-case letters indicate a significant difference between the two groups ($p < 0.05$).

with L7 (killing bacteria) via the cleavable/non-cleavable linkers and evaluated their antibacterial activity and LPS-neutralizing ability in vitro and in a mouse model of sepsis.

The cleavable/non-cleavable linkers have been widely applied in drug design to improve its bioactivity, but less attention has been paid to small molecule peptides[22,37,38]. In this work, compared to parental peptides LBP14-RKRR (+9) and L7 (+5), chimeric peptide R7 (+14) with a cleavable linker ("RKRR") displayed more potent antibacterial activity (≥ 2 ~ 4-fold increase) against the tested *E. coli*, *S. typhimurium*, and *S. aureus* (Table 2), this may be attributed to the fact that R7 with more positive charges has more potent interaction with bacterial membranes[39]; However, chimeric peptides-A7(+10) and G7(+10) based on non-cleavable linkers ("(EA₃K)₂" or "G₄S") nearly lost antibacterial activity, with the MICs of 18.1 ~ ≥ 36.2 μM, which was not consistent with previous studies[13]. This may be related to the "(EA₃K)₂" or "G₄S" linkers cannot effectively separate the independent domains of LBP14 and L7. Typically, LPS tends to form multimers with higher toxicity[40]. In the present study, the addition of R7 to the FITC-LPS micelles resulted in a more significant increase in fluorescence intensity (Fig. 4a), indicating that R7 has a stronger ability to depolymerize LPS than L7, leading to destabilization of the LPS assembly and a reduction in toxicity. This effect could be attributed to its more binding sites to LPS

than other peptides (Fig. 4b). The addition of LBP14 (targeting domain) to L7 enhanced the antibacterial and neutralizing LPS capabilities of L7, superior to the synergistic effects of LBP14 and L7 (Table 2, Figs. 4–8)[13,25]; In addition, it may be attributed to higher α-helical content of R7 (22.9 ~ 50.4%) than L7 (10.1 ~ 32.0%) in a membrane-mimic condition, resulting in its stronger antibacterial activity than the parent peptide (Supplementary Table 1 and Supplementary Table 2)[28,41,42]. In vitro, the intact L7 and LBP14-RKRR peptide products could be released from chimeric peptides R7 after cleavage with furin protease, which is beneficial to exert their corresponding functions (Supplementary Fig. 1). However, the cleavage efficiency of R7 in vitro was <36.23% within 24 h (Fig. 2a). Similar incomplete cleavage with furin in vitro also occurred in PSTCD-L2, with a 25 ~ 35% cleavage rate within 24 h, which may be due to the lack of key cellular factors of hosts[43].

To further study the cleavage efficiency of chimeric peptide R7, the mice were injected with R7 (10 ~ 50 mg/kg) by different routes, including intraperitoneal, intravenous or subcutaneous ones, respectively; blood was taken from the orbits at different time (5, 30, 60, and 120 min, respectively) and detected by RP-HPLC. Unfortunately, the expected target peaks of R7, L7 or LBP14-RKRR were not detected in mouse blood at any time point. Here are several conditions that may negatively impact RP-

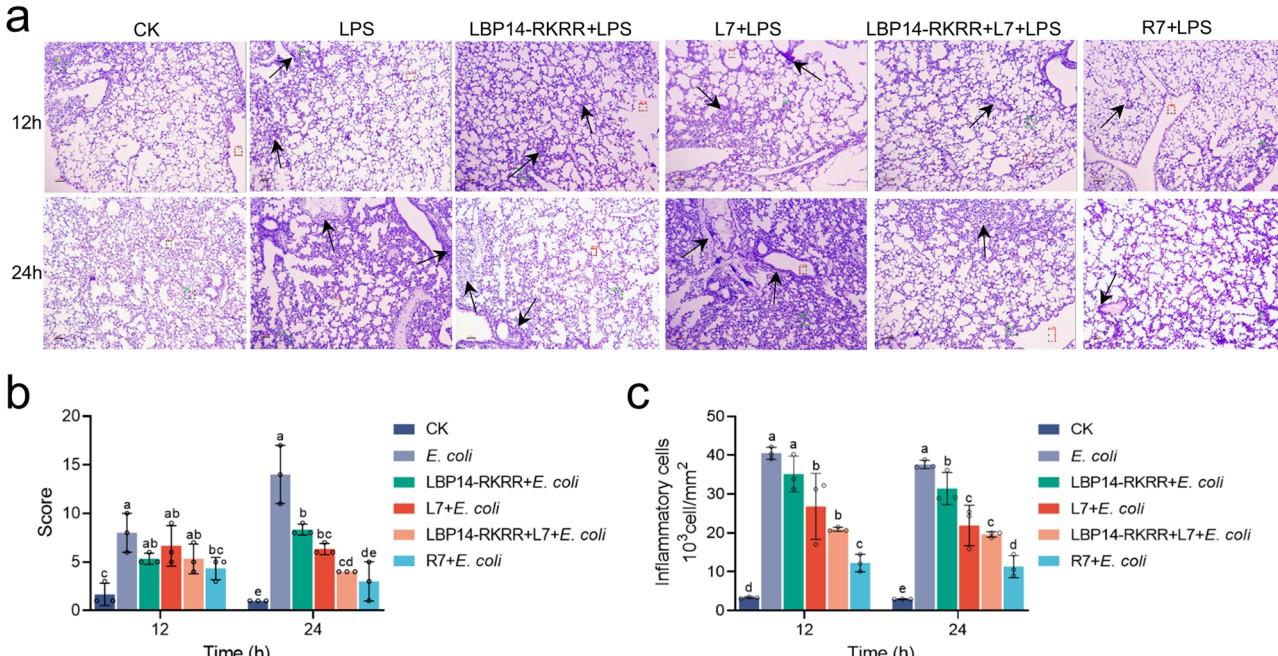

**Fig. 8 Effects of cleavable chimeric peptide R7 on lung injury in mice.** Mice were intraperitoneally injected with 7 µmol/kg different peptides (including R7, L7, LBP14-RKRR, the "LBP14-RKRR + L7 combination, respectively) at 0 h (three mice/group), followed by injection with LPS (13.5 mg/kg) at 6 h. Lung injury was analyzed histologically at 12 h or 24 h, respectively. **a** Histopathological images of lung sections after staining with hematoxylin and eosin (H&E). Images are presented at a magnification of 100×. **b** Pathological analysis of lung. Lung injury involves in inflammation (such as peribronchial, perivascular, intrabronchial, and pulmonary interstitial one), bronchial exudate, and alveolar septal thickening, respectively. The specific degree of lung injury of each mouse is assessed by Score 0 (asymptomatic), Score 1 (mild), Score 2 (moderate), and Score 3 (severe), respectively. **c** The number of counted infiltrated inflammatory cells within processed lung sections from (**a**). Data represent the mean ± SD of three independent experiments. Different lower-case letters indicate a significant difference between the two groups ($p < 0.05$).

HPLC detection results: (i) R7 may bind to other proteins such as plasma protein and form aggregates[44]; (ii) the concentration of R7 after cleavage was too low to detect target peptide by RP-HPLC; (iii) R7 may be degraded by some cellular proteases present in serum in vivo[45]. Furthermore, we used more sensitive liquid chromatograph mass spectrometer (LC-MS) to detect R7 and its possible degradation products in mice. As shown in Supplementary Table 4, a fragment "RMKKLAAPS" from the C-terminal of R7 was detected, which indicated that R7 was indeed degraded in mice. We further used MALDI-TOF MS with higher resolution to analyze R7 degradation products in blood. As shown in Supplementary Table 5, R7 was cleaved into two dual-function fragments in the serum principally, including one from position 1 to 15 that can target LPS and the other mainly from position 18 to 35 that has antibacterial activity. Notably, 6 out of 8 Arg residues in R7 especially prone to protease cleavage in mice (Supplementary Table 5), which should be replaced with non-proteolytic amino acids such as ornithine and D-Arg to stabilize peptides in blood in the next study. In addtion, in vivo stability of peptides in blood has been well modeled by ex vivo stability in serum[46]. In our study, 45.9% and 32.0% of intact R7 and L7 retained in mouse serum after incubation for 2 h; 19.7% of R7 remained in serum at 8 h, but L7 was completely undetectable in serum samples after incubation for 8 h (Fig. 3a, b), indicating that L7 may fully be degraded or bind to the serum protein[46,47] and that R7 has higher stability than L7 in serum. We also tested the degradation of R7 and L7 in human serum and found that the results were almost identical to those in mouse serum (Supplementary Fig. 2). In fact, several strategies have been proposed to improve the stability of AMPs. Substituting some enzyme-sensitive L-amino acids with D-enantiomers is the simplest way to improve the protease stability of AMPs. In previous studies,

L-amino acids in AMPs such as WLBU2, W3R, and N6NH2 were substituted with D-enantiomers[48-50], and it was found that these peptides with D-amino acids had higher stability in trypsin, simulated gastric/intestinal fluid, proteinase K, and mouse serum than the parent peptide-N6NH2. In addition, rational arrangement of amino acids is also an effective strategy. For example, Pro containing a pyrrole ring is resistant to the cleavage effect of trypsin and chymotrypsin by steric hindrance, therefore, Pro was usually placed after amino acids sensitive to protease for better stability[51]. Recently, nanotechnology has been also used to improve the protease and serum stability of AMPs. Zhou et al. designed two self-assembled peptide EN (nanofbrils structure) and NE (micelle structure), and found that self-assembly can effectively protect EN and NE from the degradation of proteases in serum[52]. It was emphasized that cleavable chimeric peptide R7 in serum remained intact within a very short time. As time increases, R7 will be partially cleaved into LBP14-RKRR and L7 and forms the mixture of R7 and "LBP14RKRR + L7" simultaneously. Finally, this mixture is gradually dominated by "LBP14-RKRR + L7" due to sufficiently cleavage of R7 (Fig. 3).

Noticeably, our studies showed that R7 does not only produce LBP14-RKRR and L7 fragments in serum, the cleavage site also occurs at other amino acids, resulting in different cleavage modes. The peptide-spectrum matches (PSMs) value represents the number of the corresponding peptides identified by the secondary spectrum in HLPC-MS. We found that the fragment with the highest PSM value was R-L7 (RFKAWRWAWRMKKLAAPS), indicating that the cleavage site occurred at the third "R" residue in the cleavable "RKRR" linker, resulting in LBP14-RKR and R-L7 segments (Supplementary Table 3 and 7). According to the PSM value, it is inferred that "LBP14-RKR + R-L7" may be more commonly seen than "LBP14-RKRR + L7". This result is not

consistent with our original design expectation. There may be certain protease in the serum that can cleave at the third "R" residue in the cleavable "RKRR" linker. Further, we synthesized LBP14-RKR and R-L7 peptides, respectively, and compared their antibacterial activity with LBP14-RKRR and L7. The results showed that the antibacterial activities of LBP14-RKR and R-L7 against the tested strains were consistent with those of LBP14-RKRR and L7, respectively (Supplementary Table 6). Therefore, R7 in the serum was principally cleaved into LPS-targeting fragment and antibacterial fragment, either "LBP14-RKRR + L7" or "LBP14-RKR + R-L7", which may exert their functions freely and it is not influenced by the other part in vivo.

A strong protection was found in mice pretreated with cleavable chimeric peptide R7 (100% survival) before MDR *E. coli* or LPS challenge, which was more effective than its parent peptide LBP14-RKRR (0% for MDR *E. coli*, 20% for LPS) and with L7 (60% for MDR *E. coli*, 40% for LPS) (Figs. 5a and 7a). Moreover, R7 could inhibit *E. coli*- or LPS-induced inflammatory response by down-regulating the production of proinflammatory factors (TNF-α, IL-6, and IL-1β) and promoting the IAP level and anti-inflammatory factor (IL-10) (Fig. 5c–g, Fig. 7b–f). Furthermore, chimeric peptide R7 protected the lungs from acute injury induced by *E. coli* or LPS and it was more effective than L7 and LBP14-RKRR (Figs. 6 and 8). Together, these results suggest that chimeric peptide R7 with the cleavable linker of "RKRR" is one of potential endotoxemia therapeutics, which may be attributed to three factors: (i) the strong ability of LBP14 in R7 to neutralize LPS (Fig. 4a); R7 may be cleaved by cellular furin in mice, followed by release of active LBP14-RKRR and L7, thereby more potently suppressing the LPS toxicity and promoting the mouse survival than L7 alone[13,53,54]; (ii) higher IAP activity induced by R7 than LBP14-RKRR and L7 (Fig. 5g and 7f), which can dephosphorylate and detoxify LPS by binding to LPS (Fig. 4) and directly altering the molecular structure of LPS, leading to the amelioration of inflammatory activity of LPS[13,16]; (iii) The R7 more promoted LPS-induced IκBα levels and reduced NF-κB p65 phosphorylation than L7 (Supplementary Fig. 5).

Noticeably, superior to other peptides (including LBP14-RKRR, L7 and their combination of 1:1 mixture), cleavage chimeric peptide R7 enhanced the survival rates in mice with sepsis, reduced lung injury and alleviated inflammatory response (Figs. 6–8). It may be related to a synergistic activity of independent functional domains of LBP14 (targeting domain) and L7 (killing domain) after cleavage in vivo, which was similar to their synergistic MIC values in vitro (Table 2). Interestingly, R7 (100% survival) exhibited a higher efficacy in the mice challenged with MDR *E. coli* and LPS than the 1:1 mixture of parental peptides "LBP14-RKRR + L7" (40 ~ 80% survival), indicating the importance of the cleavable linker of chimeric peptides. This result was similar to a previous study that P14KanS with reducible linker is more potent than the 1:1 mixture of P14LRR and kanamycin, which was used to treat the *Caenorhabditis elegans* infected with *S. enteritidis*[25].

In conclusion, we designed and characterized chimeric peptides based on LBP14 (binding to LPS) and L7 (killing bacteria) via cleavable/non-cleavable linkers. Chimeric peptide R7 with a cleavable linker could be cleaved by furin in vitro and in vivo and released L7 and LBP14-RKRR. R7 exhibited higher serum stability and antibacterial activity than its parents; A7 and G7 with the non-cleavable linkers had no antibacterial activity. R7 had a more potently ability to bind to LPS in vitro than L7. In mice, R7 more effectively protected mice from infection with MDR *E. coli* or LPS and alleviated inflammation than its parental peptides by inhibiting the production of proinflammatory cytokines (TNF-α, IL-6, and IL-1β) and promoting IAP and IL-10. It suggests that cleavable chimeric peptide R7 can be used as a potential peptide

molecule in the clinical prevention and control of bacterial infection. Furthermore, the current approach to design chimeric peptides based different linkers may provide a new perspective on developing dual-function AMPs.

## Methods

**Design and synthesis of cleavable/non-cleavable chimeric peptides.** To improve the ability of L7 to effectively neutralize LPS, three chimeric peptides are designed based on different linkers[20]. Non-cleavable chimeric peptides-A7 and G7 were designed by connecting LBP14 (LPS-targeting domain) with L7 (killing domain) by the non-cleavable flexible "G4S" and rigid "(EA3K)2" linkers. The "G4S" linker allows for flexible mobility of the two functional domains LBP14 and L7 while the "(EA3K)2" linker can maintain rigid spatial distance between LBP14 and L7. Cleavable chimeric peptide R7 was designed by connecting LBP14 with L7 by the cleavable flexible "RKRR" linker, which could be cleaved at last "R" residue of linker in vivo and released free LBP14-RKRR and L7.

The 9-fluorenylmethoxycarbonyl (Fmoc) solid-phase synthetic method is used to synthesize chimeric peptides and their parents in Mimotopes (Wuxi, China)[55]. Briefly, Fmoc-O-tert-butyl-L-serine is covalently coupled to Wang resin, and 20% piperidine is added for 20 min to remove the N-terminal Fmoc group. Subsequently, the second activated amino acid Fmoc-Pro-OH was added and coupled for 1 h, and 20% piperidine was used to remove Fmoc group again. The above process was repeated until the last amino acid was added and Fmoc was removed. The synthesized peptides were cleaved from the resin and analyzed by reversed-phase high-performance liquid chromatography (RP-HPLC) on Venusil XBP-C18 column, and their purity was over 90%. The molecular mass of peptides was confirmed by electrospray ionization mass spectrometry (ESI-MS).

**Physicochemical properties and antibacterial activities of cleavable/non-cleavable chimeric peptides.** The physicochemical properties (including molecular weight (MW), isoelectric point (pI), net charge, and aliphatic index (AI), and grand average of hydropathicity(GRAVY), respectively) of these peptides were determined using the ProtParam tool (https://web.expasy.org/protparam/). The Boman index (BI) and hydrophobicity were analyzed by Antimicrobial Peptide Calculator and Predictor (https://aps.unmc.edu/prediction/predict) and HeliQuest (https://heliquest.ipmc.cnrs.fr/cgi-bin/ComputParams.py), respectively.

The minimum inhibitory concentration (MIC) and minimum bactericidal concentration (MBC) of peptides were determined by the microtiter broth dilution method according to Clinical and Laboratory Standards Institute (CLSI) guidelines[56]. Briefly, all tested bacterial strains of *E. coli*, *Salmonella typhimurium* and *Staphylococcus aureus* were cultured to the logarithmic growth phase, diluted to $10^5$ CFU/mL, and added to the 96-well sterile plate (90 μL/well). A 2-fold dilution series of peptides in PBS (10 μL) were added to the plates to give the final concentrations of 0.5, 1, 2, 4, 8, 16, 32, and 64 μM, respectively. For "LBP14-RKRR + L7" and "LBP14 + L7", the same molar amount (equal to that of R7) of parent peptides were mixed first, and then diluted by gradient. PBS was used as the negative control. The cell plate was incubated at 37 °C for 16 ~ 18 h until a visible turbidity was observed in the negative control. The lowest concentration that completely inhibited bacterial growth was the MIC value of peptides against the tested strains. The MBC value was calculated as the lowest concentration of peptides resulting in a 99.9% reduction of the original inoculum. All experiments were repeated three times.

**Serum stability, hemolysis and cytotoxicity of cleavable chimeric peptide R7**. Serum stability of R7 and its parent-L7 was tested as previously described with minor modifications[46,53]. Briefly, each peptide was added to the 25% mouse serum to a final concentration of 0.5 mM; 250 μL aliquots of samples were taken at 0, 0.5, 1, 2, 4 and 8 h, and mixed with 20 μL 15% trichloroacetic acid (TCA). The samples were incubated at 4 °C for 15 min and centrifuged for 10 min at 12,000 rpm. The supernatants were analyzed by RP-HPLC or inhibition zone method. Moreover, after incubation in serum for 2 h, R7 was analyzed by matrix assisted laser desorption/ionization-time of flight mass spectrometry (MALDI-TOF MS) and high-performance liquid chromatography-mass spectrometry (HPLC-MS), respectively.

Blood was collected from the eyeballs of specific pathogen-free (SPF) ICR female mice (6-week-old) with an anticoagulant tube and centrifuged for 10 min (4 °C, 1500 rpm). After washing twice with 0.9% physiological saline, red blood cells were diluted to 8% suspension and added into the Eppendorf tube (100 μL); 2-fold peptide solutions dissolved in 0.9% physiological saline (100 μL) were then added into the tube, incubated at 37 °C for 1 h, and centrifugated at 4 °C for 5 min. The supernatant was pipetted into a 96-well plate and the UV absorbance was measured at 540 nm using a microplate reader. 0.1% Triton X-100 and physiological saline were used as positive and negative control, respectively. The hemolysis rate is calculated as follows: Hemolysis (%) = [(Abs$_{peptide}$-Abs$_{saline}$) / (Abs$_{0.1\%\ Triton\ X-100}$-Abs$_{saline}$)] × 100[13].

The 3-(4, 5-dimethylthiazolyl-2)-2, 5-diphenyltetrazolium bromide (MTT) method was used to detect the cytotoxicity of R7 and L7. RAW264.7 cells ($2.5 \times 10^4$ cells/mL, 100 μL) were added into the 96-well plate and cultured at 37 °C in 5% $CO_2$ for 24 h. After removing the medium and washing twice with PBS, 100 μL peptides (1, 2, 4, 8, 16, 32, 64 and 128 μM) were added to each well. PBS was served as the negative control. After incubation for 24 h, 20 μL MTT (5 mg/mL) was added into each well and incubated for another 4 h. Then, the supernatant was removed and 150 μL dimethylsulfoxide (DMSO) was added into the plate and shaken for 10 min. The absorbance of each well was measured at 570 nm and the survival rate of cells was calculated as follows: Survival (%) = [(Abs$_{peptide}$/ Abs$_{control}$)] × 100, where Abs$_{peptide}$ and Abs$_{control}$ present the absorbance of peptide and PBS control, respectively.

**Circular dichroism (CD) analysis of cleavable chimeric peptide R7**. The secondary structure of chimeric peptide R7 and its parent-L7 was detected in ddH$_2$O, sodium dodecyl sulfate (SDS), and trifluoroethanol (TFE) buffer by CD spectroscopy, respectively. The CD spectra was performed as described previously[57]. In brief, R7 was dissolved in ddH$_2$O, SDS (5, 20, and 40 mM), and 50% TFE, respectively. A total of 200 μL mixture was added to a quartz cuvette (0.1 cm path length) and spectra were recorded from 190 to 260 nm at 25 °C. CDNN software is used to analyze the secondary structure of peptides.

**Release of L7 and LBP14 from cleavable chimeric peptide R7 in vitro**. Chimeric peptide R7 was treated with recombinant human furin enzyme (8 U/reaction; NEB P8077) in 50-μL reactions supplemented with in 20 mM HEPES, 0.1% Triton X-100, 1 mM CaCl$_2$, 0.2 mM β-mercaptoethanol. The reaction mixtures were incubated at 25 °C for 65 h. An aliquot of 50 μL sample was taken at different time points, and analyzed by RP-HPLC or MALDI-TOF MS. L7 or LBP14-RKRR were used as control, respectively. The cleavage rate of R7 was calculated from the peak area as follows: Cleavage rate (%) = [(Area$_0$ −Area$_t$)/Area$_0$] × 100, where 0 and t present digestion time[58].

**Interaction between cleavable chimeric peptide R7 and *E. coli* LPS**. The binding of peptides to LPS was determined according to the method described previously[34,59]. In brief, *E. coli* 0111: B4 LPS (40 μg/mL) and BODIPY-TR-cadaverine (BC) (10 μM) were mixed in 50 mM Tris buffer (pH = 7.4) in equal proportions and added to a 96-well black plate (180 μL/well) in the dark. Different concentrations of chimeric peptide R7 and its parental (20 μL) were added into each well and detected by a fluorescence spectrophotometry at room temperature (excitation wavelength 580 nm, emission wavelength 620 nm). The change in relative fluorescence was recorded and the BC displacement rate was calculated. Polymycin B (PMB) was used as the positive control.

Docking of ligands (R7 and L7) were performed on the structure of receptor (LPS) using AutoDock Vina as the previous method with some modifications[60]. The structures of peptides were obtained from the AlphaFold Colab (https://colab.research.google.com/github/deepmind/alphafold/blob/main/notebooks/AlphaFold.ipynb). The grid box of LPS was $110 \times 110 \times 110$ points, and the grid space was 0.375 Å. All molecules keep rigid, and the final complex conformation that had the minimum binding energy was chosen as the putative binding site. The results were displayed by PyMol1.8.

**Efficacy of cleavable chimeric peptide R7 and its parents in mice**. The SPF ICR female mice (6-week-old) were purchased from the Beijing Vital River Laboratory Animal Technology Co. Ltd. The animal experiments were undertaken with the permission of the Animal Care and Use Committee, Institute of Feed Research, Chinese Academy of Agricultural Sciences. We have complied with all relevant ethical regulations for animal use.

A sepsis mouse model established in our laboratory was used to evaluate the prevention effect of peptides[13]. As the previous method of intraperitoneal injection[61], the left abdominal cavity of mice (five mice/group) was injected with 7 μmol/kg (minimum effect dose in preliminary test (Supplementary Fig. 4)) chimeric peptide-R7, L7, LBP14-RKRR, the "LBP14-RKRR + L7" combination or PBS, followed by the right intraperitoneal injection with a lethal dose of MDR *E. coli* CVCC195 or LPS (13.5 mg/kg) after 6 h. The survival rate of mice was recorded for 7 days. The mice injected with saline were used as negative control.

After *E. coli* or LPS challenge, the mice were sacrificed at 12 h and 24 h, respectively, and the sera were collected to detect inflammatory cytokines (such as tumor necrosis factor-α (TNF-α), interleukin-6 (IL-6), interleukin-1β (IL-1β), and interleukin-10 (IL-10)) using the enzyme linked immunosorbent assay (ELISA) kits. To evaluate the specific activity of intestinal alkaline phosphatase (IAP), a 2 cm segment of duodenum was rapidly intercepted from the mice and homogenized; the IAP levels in the supernatant were detected by ELISA. In addition, the lungs were collected from the mice infected by *E. coli* after 12 and 24 h, respectively and were homogenized in sterile PBS for the bacterial counts assay.

Lung tissues were taken at 12 h and 24 h, respectively, washed with PBS, and fixed in 4% paraformaldehyde overnight. After dehydration in different concentrations of ethanol (including 75%, 85%, 90%, and 95%, respectively), the tissues were embedded in paraffin, sliced, and stained with hematoxylin. Observation was performed under a light microscope. The mice only injected with *E. coli*, LPS or PBS was regard as the negative or blank control.

Additionally, the lung tissues of 24 h were homogenized and suspended in 5 M HEPES buffer, and the protein levels of supernatants were quantified using a Bradford protein assay kit. A total of 30 μg proteins were analyzed on a sodium dodecyl

sulfate-polyacrylamide gel electrophoresis gel and were immuno-blotted onto PVDF membranes, followed by an incubation with the primary antibodies of p65, p-p65, extracellular signal-regulated kinase (ERK)1/2, p-ERK1/2 and IκBα overnight at 4 °C. After washing, the blots were incubated with the peroxidase conjugated goat-rabbit or goat-mouse antibodies. Relative protein expression levels were quantified by a densitometric measurement of chemiluminescence (ECL) reaction band. β-actin was used as the internal reference.

**Statistics and Reproducibility.** Unless otherwise noted, experiments were repeated at least three times. All data are presented as means ± standard deviation (SD). Multiple comparisons between groups were analyzed by R studio and different lower-case letters indicate a markedly difference between two groups ($p < 0.05$).

**Reporting summary.** Further information on research design is available in the Nature Portfolio Reporting Summary linked to this article.

## Data availability
The relevant data are available from the corresponding authors upon request. The source data underlying Figs. 1a–d, 2b, 3b–d, 4a, 5a–g, 6b, c, 7a–f, 8b, c, supplementary Figs. 2b–d, 3a–b, 4, and 5b–d are provided as a Supplementary Data files.

## Code availability
Custom codes that support the findings of this study are available from the first author upon reasonable request.

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

## Acknowledgements

This work was supported by the National Natural Science Foundation of China (grants No. 31772640, No. 31572444, and No. 31572445), the AMP Direction of National Innovation Program of Agricultural Science and Technology from Chinese Academy of Agricultural Sciences, China (grant No. CAAS-ASTIP-2013-FRI-02), and its Key Project of Alternatives to Antibiotic for Feed Usages from Chinese Academy of Agricultural Sciences, China (grant No. CAAS-ZDXT2018008). We appreciate the help of Lili Niu from the Institute of Biophysics at the Chinese Academy of Sciences (CAS) for valuable analysis with the MALDI-TOF mass spectrometry.

## Author contributions

J.W., X.W., and Z.W. conceived and designed the experiments; Z.L.W. conducted the experiments; D.T., R.M., Y.H., and N.Y. supervised the work, provided some data analysis and revised the final version of the manuscript; Z.W., X.W., and J.W. wrote the manuscript.

## Competing interests

The authors declare no competing interests.
