## [Peer Review File · Communications Biology]

Reviewers' comments:

Reviewer #1 (Remarks to the Author):

The present paper describes the chimeric peptide-R7, which was generated by assembling the sequences of the LPS-binding peptide LBP14 and the weakly antimicrobial peptide L7 via a cleavable "RKRR" linker into R4 displaying improved antibacterial and anti-inflammatory activities.

Even though the activities of R7 are very interesting the manuscript in its current form suffers from two major weaknesses that need to be addressed prior to acceptance:

1) The English is far from the standard of scientific publications - the entire manuscript requires language polishing (just a few examples of errors and/or poor grammar are indicated in the attached annotated pdf)

2) The main conclusion repeated several places in the manuscript that the cleavable linker is responsible for the improved properties on the conjugated peptide appears not very likely - all data indicate that the beneficial activities arise from the combination of these moieties into a more stable and more active peptide. Thus R7 is more potent than L7 and LBP-14 with respect to both antibacterial and LPS-binding properties (Fig. 5A), and hence the experimental data are not convincingly supporting the advantage of the cleavable linker.

Additional minor issues is shown as annotations in the pdf enclosed.

Reviewer #2 (Remarks to the Author):

In this manuscript, the authors report design, synthesis, *in vitro* and *in vivo* evaluation of a chimeric peptide as an antibacterial agent against gram-negative MDR E. coli. The chimeric peptide consists of an L7 antibacterial peptide and an LBP14 LPS-binding peptide linked via 4-mer furin (protease) cleavable peptide. The goal is to come up with a new peptide (R7) with improved antibacterial and anti-inflammatory properties.

In general, the manuscript is well written, and the steps for designing, synthesizing, and evaluating *in vitro/in vivo* activity are described appropriately. I believe this manuscript will be of interest to a large portion of the readers of this journal. I recommend this study for publication after addressing all the comments below.

Major Comments:

1. Peptide stability (half-life) is most important when evaluating for *in vivo* efficacy. The authors find that after injecting R7 peptide via IP, IV or SQ there was no peptide detected in blood of mice. The authors mention several reasons for this finding, but this is expected result from a non-stabilized peptide. Most L-peptides will have a half-life of 15 minutes or even less when injected. This is the main issue with this study, and authors should comment on it. Some strategies to stabilize the peptide in the future could be proposed.

2. The *in vitro* stability was studied in 25% bovine serum. It should be explained why human serum was not used instead.

3. Fig 6A and 7A. There were 7 groups used for the efficacy study. However, it is not mentioned how many mice were in each treatment group. As presented, it appears there were 3 mice per group. This should be clearly mentioned as n=3. Also, the average from three mice should be presented in figures 6A/7A. Also, are three mice per group going to give statistically significant data?

Other comments:

1. Mice were injected with 7 micromol/kg peptide. How was this dose decided must be mentioned?
2. Figure 2 legend. Mentions "CD spectra of L7 and R7". This should be revised as the CD only shows for R7.

Reviewer #3 (Remarks to the Author):

Overall:

The introduction is comprehensive and concise. The logic and experimental aims are clearly laid out. The idea of linking these two domains (an LPS binding domain and antimicrobial peptide domain) is interesting, especially when using a cleavable linker. The results point towards a hybrid peptide that is especially good at binding LPS.

I think the paper is overall of good quality, showing novel results and enough evidence for the claims that are made. The linked peptide shows promise as anti-infective (or anti-septic) drug, although clinical application is still far away. The paper is sometimes slightly hard to read, as number pile up in sentences in the results section.

Minor comments:

A definition of R7 is missing in the introduction, which might confuse the reader further on.

A small comment on the use of MIC: while fairly standardized, the assay does not give information of the viability of the bacteria, merely on the growth inhibition demonstrated by the peptides.

Determining the minimal bactericidal concentration (MBC) would be more indicative of actual killing activity.

PMB is not defined in table 1.

In figure 3, there is a third peak that starts to become more visible after 18h that is not annotated.

Figure 4: in this figure, the authors show in 4B that after 8h only 20% of the peptide is remaining.

However, in 4C, the authors suggest that this has no effect on the activity of the peptide. I think an experimental setup more in line with their previous MIC determinations would be more appropriate.

Figure 6 should indicate in the legend how many mice per group were used. Additionally, there seems to be no significant difference between R7 and L7 + LBP-RKRR. In further panel in the figure, the comparison between L7 and R7 is shown; I think L7 + LBP-RKRR and R7 should also be included.

Response to reviewers

Reviewers' comments:

Reviewer #1 (Remarks to the Author):

Question 1: The English is far from the standard of scientific publications - the entire manuscript requires language polishing (just a few examples of errors and/or poor grammar are indicated in the attached annotated pdf)

Response 1: Thank you for your comments. We have carefully checked and improved the English writing, and the minor issues annotated in the PDF, including spelling and grammar errors have been corrected in the revised manuscript.

Question 2: The main conclusion repeated several places in the manuscript that the cleavable linker is responsible for the improved properties on the conjugated peptide appears not very likely - all data indicate that the beneficial activities arise from the combination of these moieties into a more stable and more active peptide. Thus R7 is more potent than L7 and LBP-14 with respect to both antibacterial and LPS-binding properties (Fig. 5A), and hence the experimental data are not convincingly supporting the advantage of the cleavable linker. Additional minor issues is shown as annotations in the pdf enclosed.

Response 2: Actually, cleavable chimeric peptide R7 *in vivo* remained intact within a very short time. As time increases, R7 was partially cleaved into LPS-targeting and bactericidal fragments (LBP14-RKRR and L7), and formed the mixture of R7 and "LBP14RKRR+L7" simultaneously. Finally, this mixture was gradually dominated by "LBP14-RKRR +L7" due to sufficiently cleavage of R7.

Additionally, after cleavage of R7, the "LBP14RKRR+L7" mixture was more potent than the cocktail (1:1 mixture of LBP14-RKRR and L7) in mice (Fig. 5a, Fig. 7a), indicating the importance of the cleavable linker of chimeric peptides. This result was similar to a previous study that P14KanS with a cleavable linker was more potent than the 1:1 mixture of P14LRR and kanamycin, which was used to treat the *Caenorhabditis elegans* infected with *S. enteritidis* (Brezden, et al., 2016). Thus, improved activity of chimeric peptides may be not from the simple combination of their parental moieties, which needs a further study in our next work.

Fig. 5. Efficacy of cleavable chimeric peptide R7 in the mice challenged with MDR *E. coli*. a Survival of

mice. Mice were intraperitoneally injected with 7 $\mu\text{mol/kg}$ different peptides (including R7, L7, LBP14-RKRR, the "LBP14-RKRR+L7 combination") at 0 h (five mice/group), followed by injection with MDR *E. coli* CVCC195 (0.5×10^9 CFU/mouse) at 6 h. Survival was recorded for 7 d. Two biological replicates were done.

Fig. 7. Efficacy of cleavable chimeric peptide R7 in the mice challenged with LPS. a Survival of mice. Mice were intraperitoneally injected with different peptides (7 $\mu\text{mol/kg}$) at 0 h (five mice/group), followed by injection with LPS (13.5 mg/kg of body weight) at 6 h. Survival was recorded for 7 d. Two biological replicates were done.

Reviewer #2 (Remarks to the Author):

Question 1: Peptide stability (half-life) is most important when evaluating for in vivo efficacy. The authors find that after injecting R7 peptide via IP, IV or SQ there was no peptide detected in blood of mice. The authors mention several reasons for this finding, but this is expected result from a non-stabilized peptide. Most L-peptides will have a half-life of 15 minutes or even less when injected. This is the main issue with this study, and authors should comment on it. Some strategies to stabilize the peptide in the future could be proposed.

Response 1: Yes, you are right. We have added a comment regarding strategies to stabilize the peptide in the Discussion section.

Pages 27-28, Line 428-440: "In fact, several strategies have been proposed to improve the stability of AMPs. Substituting some enzyme-sensitive L-amino acids with D-enantiomers is the simplest way to improve the protease stability of AMPs. In previous studies, L-amino acids in AMPs such as WLBU2, W3R, and N6NH₂ were substituted with D-enantiomers (Di et al., 2020; Li et al., 2019; Li, et al., 2020), and it was found that these peptides with D-amino acids had higher stability in trypsin, simulated gastric/intestinal fluid, proteinase K, and mouse serum than the parent peptides. In addition, rational arrangement of amino acids is also an effective strategy. For example, Pro containing a pyrrole ring is resistant to trypsin and chymotrypsin by steric hindrance; therefore, Pro is usually placed after amino acids sensitive to protease for better stability (Zhu et al., 2020)." Recently, nanotechnology has been also used to improve the protease and serum stability of AMPs. Zhou et al. designed two self-assembled peptide EN (nanofibrils structure) and NE (micelle structure), and found that self-assembly can effectively p

protect EN and NE from the degradation of proteases in serum (Zhou et al., 2017).

Question 2: The in vitro stability was studied in 25% bovine serum. It should be explained why human serum was not used instead.

Response 2: Thanks for your suggestion. Our aim is that these peptides may be firstly used in animals because our study mainly focus on the poultry and livestock. Therefore, in this study, we firstly used bovine serum to study the stability of peptides. In this revision, we chose mouse serum to further validate our results; it was found that the stability of the peptides in mouse serum (Fig. 3) was similar to that in bovine serum (Fig. S2). Additionally, the stability of peptides in human serum may be performed in our further study.

Fig. 3. Stability and release of peptides in mouse serum. (a) Detection of the peak area change of chimeric peptide R7 and its parent L7 in serum by RP-HPLC. (b) The remaining rate of R7 and L7 in mouse serum. (c) The antibacterial activity of R7 and L7 against MDR *E. coli* in mouse serum detected by an inhibition assay. (d) The MALDI-TOF MS of R7 after incubation in mouse serum for 2 h. The experiments were repeated three times. Data in (a) and (d) is representative of three biological replicates. The results are given as the mean \pm SD of three independent experiments in (b) and (c). Different lower-case letters indicate a significant difference between the two groups ($p < 0.05$).

Fig. S2. Serum stability and release of peptides in bovine serum. (a) Detection of the peak area change of peptides in serum by RP-HPLC. (b) The remaining rate of peptides in bovine serum. (c) The antibacterial activity of peptides against MDR *E. coli* in serum detected by an inhibition assay. (d) The MALDI-TOF mass spectrometry of R7 after incubation in bovine serum for 2 h. The experiments were repeated three times. Data in (a) and (d) is representative of three biological replicates. The results are given as the mean \pm SD of three independent experiments in (b) and (c). Different lower-case letters indicate a significant difference between the two groups ($p < 0.05$).

Question 3: Fig 6A and 7A. There were 7 groups used for the efficacy study. However, it is not mentioned how many mice were in each treatment group. As presented, it appears there were 3 mice per group. This should be clearly mentioned as $n=3$. Also, the average from three mice should be presented in figures 6A/7A. Also, are three mice per group going to give statistically significant data?

Response 3: In Fig. 6A and 7A, there were 5 mice in each group, and we performed two replicate experiments at different times with the same results. According to your suggestion, we added the “n” in each group in the method and legend in the revised manuscript as follow:

Page 35, Line 600: “Briefly, the mice (five mice/group) were intraperitoneally injected with 7 $\mu\text{mol/kg}$ peptides (R7, L7, LBP14-RKRR, and LBP14-RKRR+L7) and challenged with a lethal dose of MDR *E. coli* CVCC195 (0.5×10^9 CFU/mouse) or LPS (13.5 mg/kg) after 6 h administration.” (Page 35)

Page 19, Lines 274-276: “Fig. 5. Efficacy of cleavable chimeric peptide R7 in the mice challenged with MDR *E. coli*. (a) Survival of mice. Mice were intraperitoneally injected with 7 $\mu\text{mol/kg}$ different peptides (including R7, L7, LBP14-RKRR, the “LBP14-RKRR+L7 combination) at 0 h (five mice/group)”.

Page 24, Lines 353-354: “Fig. 7. Efficacy of cleavable chimeric peptide R7 in the mice challenged with LPS. (a) Survival of mice. Mice were intraperitoneally injected with different peptides (7 $\mu\text{mol/kg}$) at 0 h (five mice/group)”

Question 4: Mice were injected with 7 micromol/kg peptide. How was this dose decided must be mentioned?

Response 4: This dose of 7 micromol/kg was the minimum effect dose in preliminary test. The results of test have been added in Supplementary materials as Fig. S3

Fig. S3. Efficacy of different doses of cleavable chimeric peptide R7 in the mice challenged with MDR *E. coli*. Mice were intraperitoneally injected with 3.5, 7 or 14 $\mu\text{mol/kg}$ R7 at 0 h (five mice/group), followed by injection with MDR *E. coli* CVCC195 (0.5×10^9 CFU/mouse) at 6 h. PBS was used as control. Survival was recorded for 7 d.

Question 5. Figure 2 legend. Mentions “CD spectra of L7 and R7”. This should be revised as the CD only shows for R7.

Response 5: Thank your suggestion. In the original manuscript, only the CD spectrum of R7 was detected. In this revision, we supplemented the CD spectra of L7 to investigate the structural features between L7 and R7.

Page11, Lines 149-156: “The secondary structures of L7 and R7 in ddH₂O were characterized predominantly by antiparallel, β -turn and random coils with a characteristic positive maximum at 230 nm and a negative minimum at 200 nm (Fig. 1c and d). In SDS and 50% TFE buffer, chimeric peptide R7 exhibited a significant increase in α -helical content (22.9~50.4%) with a characteristic positive peak at 190 nm and two negative peaks at 208 and 222 nm, respectively, higher than the parent peptide L7 (10.1~32.0%). Particularly, α -helix of R7 in 50% TFE was up to 50.4% higher than L7 (13.9%) (Table S1 and S2). These results suggest that R7 is more prone to form α -helix conformation in cell membrane environment.”

Fig. 1. The CD spectra of cleavable chimeric peptide R7 and its parent L7. (a) Hemolytic activity of R7 and L7 against mouse erythrocyte. (b) The cytotoxic activity of R7 and L7 against RAW 264.7 monocytes by MTT assay. CD spectra of L7 (c) and R7 (d) in water, SDS, and 50% TFE, respectively. Results indicate means with SD (n = 3 independent experiments).

Table S1 CD analysis of secondary structures of L7 in water, SDS, and 50% TFE.

Secondary structures	The percentages of secondary structures (%) of L7				
	H ₂ O	5 mM SDS	20 mM SDS	40 mM SDS	50% TFE
α -Helix	4.8	32.0	10.2	10.1	13.9
Antiparallel	26.3	11.1	41.6	41.4	42.0
Parallel	2.6	9.2	4.0	4.0	5.1
β -Turn	25.4	17.8	19.9	19.9	18.3
Random coil	40.8	30.0	24.4	24.6	20.7

Table S2 CD analysis of secondary structures of R7 in water, SDS, and 50% TFE.

Secondary structures	The percentages of secondary structures (%) of R7				
	H ₂ O	5 mM SDS	20 mM SDS	40 mM SDS	50% TFE
α -Helix	3.5	35.4	22.9	23.5	50.4
Antiparallel	10.1	8.5	26.8	25.4	0.8
Parallel	1.7	9.8	6.5	6.5	8.7
β -Turn	31.2	17.5	20.9	21.1	16.4
Random coil	53.5	28.8	23.0	23.5	23.7

Reviewer #3 (Remarks to the Author):

Question 1: The paper is sometimes slightly hard to read, as number pile up in sentences in the results section.

Response 1: The unimportant number was deleted from the results section.

Question 2: A definition of R7 is missing in the introduction, which might confuse the reader further on.

Response 2: Thank you, we rewritten the “Introduction” part and defined the R7 as follow:

Page 5, Line 85-88: “In this study, to improve their biological activity, cleavable and non-cleavable chimeric peptides- R7, A7 and G7 were designed by connecting LBP14 (targeting domain) with L7 (killing domain) via the cleavable (“RKRR”) and non-cleavable (“(EA₃K)₂” or “G₄S”) linkers, respectively.”

Question 3: A small comment on the use of MIC: while fairly standardized, the assay does not give information of the viability of the bacteria, merely on the growth inhibition demonstrated by the peptides. Determining the minimal bactericidal concentration (MBC) would be more indicative of actual killing activity. PMB is not defined in table 1.

Response 3: i) The method of MIC and MBC of R7 and other peptides were improved and added in this revision.

Page 32, Lines 525-535: “The minimum inhibitory concentration (MIC) and minimum bactericidal concentration (MBC) of peptides were determined by the microtiter broth dilution method according to Clinical and Laboratory Standards Institute (CLSI) guidelines (CLSI, 2009). ...The lowest concentration that completely inhibited bacterial growth was the MIC value of peptides against the tested strains. The MBC value was calculated as the lowest concentration of peptides resulting in a 99.9% reduction of the original inoculum.”

ii) We re-measured MIC and MBC for all peptides, and also gave the full name of PMB in above Table 2.

Table 2 MIC and MBC values of chimeric peptide and its parents.

Strains	LBP14		LBP14-RKRR		L7		R7		LBP14 + L7 ^a		LBP14-RKRR + L7 ^a		Polymycin B (PMB)		A7 ^b		G7 ^c	
	MIC (μM)	MBC (μM)	MIC (μM)	MBC (μM)	MIC (μM)	MBC (μM)	MIC (μM)	MBC (μM)	MIC (μM)	MBC (μM)	MIC (μM)	MBC (μM)	MIC (μM)	MBC (μM)	MIC (μM)	MBC (μM)	MIC (μM)	MBC (μM)
Gram-negative bacteria																		
Escherichia coli CVCC195 ^d	64	NT ^e	32	32	32	64	16	16	16	NT	16	32	1	1	128	128	64	64
S. typhimurium CVCC533	>64	NT	64	128	>128	>128	16	32	64	NT	64	64	8	8	>128	>128	>128	>128
Gram-positive bacteria																		
Staphylococcus aureus ATCC43300 ^d	>64	NT	64	64	>128	>128	32	32	>64	NT	32	32	>64	>64	>128	>128	>128	>128
S. aureus CVCC546 ^d	>64	NT	128	128	128	>128	32	32	64	NT	32	32	>64	>64	>128	>128	>128	>128

^a 1:1 mixture of LBP14 (or LBP14-RKRR) and L7. ^b A7: connecting LBP14 and L7 via a rigid linker "(EA₃K)₂". ^c G7: connecting LBP14 and L7 via a flexible linker

"G₄S". ^d The strains are multi-drug resistant to different antibiotics^{9,42}. ^e NT: not tested.

Question 4: In figure 3, there is a third peak that starts to become more visible after 18h that is not annotated.

Response 4: Thanks, we marked all visible peaks in this revision.

Fig. 2. Release of L7 from cleavable chimeric peptide R7 *in vitro*.

Question 5: Figure 4: in this figure, the authors show in 4B that after 8h only 20% of the peptide is remaining. However, in 4C, the authors suggest that this has no effect on the activity of the peptide. I think an experimental setup more in line with their previous MIC determinations would be more appropriate.

Response 5: In fact, compared to that at 0 h, approximately 20% of intact R7 was detected at 8 h by HPLC (ratio of peak areas at 8 h and 0 h); 80% R7 may be present in other ways, such as "LBP14-RKRR+L7". According to Table 2, R7 and "LBP14-RKRR+L7" displayed same antibacterial activity against *E. coli*; thus, effects of 20% R7 and 80% "LBP14-RKRR+L7" were equivalent to that of 100% R7, indicating similar activity.

Question 6: Figure 6 should indicate in the legend how many mice per group were used. Additionally, there seems to be no significant difference between R7 and L7 + LBP-RKRR. In further panel in the figure, the comparison between L7 and R7 is shown; I think L7 + LBP-RKRR and R7 should also be included.

Response 6: In Fig. 6 and 7, there were 5 mice in each group, and we defined the "n" in each group in this revision. Additionally, we re-performed *in vivo* experiments and added the two "LBP14-RKRR" and "LBP14-RKRR+L7" groups as controls. Cytokines and lung tissue damage were also evaluated in all groups (Fig.5-8).

Fig. 5. Efficacy of cleavable chimeric peptide R7 in the mice challenged with MDR *E. coli*. **a** Survival of mice. Mice were intraperitoneally injected with 7 $\mu\text{mol/kg}$ different peptides (including R7, L7, LBP14-RKRR, the “LBP14-RKRR+L7 combination”) at 0 h (five mice/group), followed by injection with MDR *E. coli* CVCC195 (0.5×10^9 CFU/mouse) at 6 h. Survival was recorded for 7 d. Two biological replicates were done. **b** Effects of peptides on the bacterial counts in lungs. Mice were infected intraperitoneally with MDR *E. coli* CVCC195 (0.5×10^9 CFU/mouse) and pre-treated with 7 $\mu\text{mol/kg}$ different peptides at 6 h (three mice/group) before infection, respectively. Lungs were removed at 12 h and 24 h after infection to analyze bacterial translocation. **c-g** Effects of peptides on cytokines (**c-f**) and the IAP levels in mice (**g**). The results are given as the mean \pm SD of three independent experiments. Different lower-case letters indicate a significant difference between the two groups ($p < 0.05$).

Fig. 6. Effects of cleavable chimeric peptide R7 on lung injury in mice. Mice were intraperitoneally injected with 7 μ mol/kg different peptides (including R7, L7, LBP14-RKRR, and the “LBP14-RKRR+L7 combination, respectively) at 0 h (three mice/group), followed by injection with MDR *E. coli* CVCC195 (0.5×10^9 CFU/mouse) at 6 h. Lung injury was analyzed histologically at 12 h or 24 h. **a** Histopathological images of lung sections after staining with hematoxylin and eosin (H&E). Images are presented at a magnification of 100 \times . **b** Pathological analysis of lung. Lung injury involves inflammation (such as peribronchial, perivascular, intrabronchial, and pulmonary interstitial one), bronchial exudate, and alveolar septal thickening, respectively. The specific degree of lung injury of each mouse is assessed by Score 0 (asymptomatic), Score 1 (mild), Score 2 (moderate), and Score 3 (severe), respectively. **c** The number of counted infiltrated inflammatory cells within processed lung sections from **(a)**. Data represent the mean \pm SD of three independent experiments. Different lower-case letters indicate a significant difference between the two groups ($p < 0.05$).

Fig. 7. Efficacy of cleavable chimeric peptide R7 in the mice challenged with LPS. **a** Survival of mice. Mice were intraperitoneally injected with different peptides (7 $\mu\text{mol/kg}$) at 0 h (five mice/group), followed by injection with LPS (13.5 mg/kg of body weight) at 6 h. Survival was recorded for 7 d. Two biological replicates were done. **b-f** Effects of L7 and R7 on cytokines (**b-e**) and the IAP levels (**f**). The results are given as the mean \pm SD of three independent experiments. Different lower-case letters indicate a significant difference between the two groups ($p < 0.05$).

Fig. 8. Effects of cleavable chimeric peptide R7 on lung injury in mice. Mice were intraperitoneally injected with 7 $\mu\text{mol/kg}$ different peptides (including R7, L7, LBP14-RKRR, the "LBP14-RKRR+L7 combination, respectively) at 0 h (three mice/group), followed by injection with LPS (13.5 mg/kg) at 6 h. Lung injury was analyzed histologically at 12 h or 24 h, respectively. **a** Histopathological images of lung sections after staining with hematoxylin and eosin (H&E). Images are presented at a magnification of 100 \times . **b** Pathological analysis of lung. Lung injury involves inflammation (such as peribronchial, perivascular, intrabronchial, and pulmonary interstitial one), bronchial exudate, and alveolar septal thickening, respectively. The specific degree of lung injury of each mouse is assessed by Score 0 (asymptomatic), Score 1 (mild), Score 2 (moderate), and Score 3 (severe), respectively. **c** The number of counted infiltrated inflammatory cells within processed lung sections from (**a**). Data represent the mean \pm SD of three independent experiments. Different lower-case letters indicate a significant difference between the two groups ($p < 0.05$).

Reference:

- Brezden, Anna et al. "Dual targeting of intracellular pathogenic bacteria with a cleavable conjugate of kanamycin and an antibacterial cell-penetrating peptide." *Journal of the American Chemical Society* vol. 138, 34 (2016): 10945-9.
- Di, Y P et al. "Enhanced therapeutic index of an antimicrobial peptide in mice by increasing safety and activity against multidrug-resistant bacteria." *Science advances* vol. 6, 18 eaay6817. 1 May. 2020
- Li, Yi et al. "Antimicrobial activity, membrane interaction and stability of the D-amino acid substituted analogs of antimicrobial peptide W3R6." *Journal of photochemistry and photobiology. B, Biology* vol. 200 (2019): 111645.
- Li, Ting et al. "Dual antibacterial activities and biofilm eradication of a marine peptide-N6NH2 and its analogs against multidrug-resistant *Aeromonas veronii*." *International journal of molecular sciences* vol. 21, 24 9637. 17 Dec. 2020
- Zhu, Yongjie et al. "Rational avoidance of protease cleavage sites and symmetrical end-tagging significantly enhances the stability and therapeutic potential of antimicrobial peptides." *Journal of medicinal chemistry* vol. 63, 17 (2020): 9421-9435.
- Zhou, Xi-Rui et al. "Self-assembly nanostructure controlled sustained release, activity and stability of peptide drugs." *International journal of pharmaceutics* vol. 528,1-2 (2017): 723-731.
- Clinical & Institute, L. S. Performance standards for antimicrobial susceptibility testing; Nineteenth th Informational Supplement. M100-S19 (2009).

Reviewers' comments:

Reviewer #3 (Remarks to the Author):

The manuscript has been greatly improved. I have no further additions.

Reviewer #4 (Remarks to the Author):

The authors have revised the manuscript according to the original comments of which most have been done satisfactorily.

I have however one major concern related to the comments on stability and the general observation that it is very hard to link the in vitro studies to the in vivo outcome.

In detail:

The chimeric R7 peptide or any of its parent molecules cannot be detected in blood. This is likely due to the short half life in blood. Its attachment or uptake to blood cells or serum components. It is hard to imagine how direct antibacterial activity can then be accomplished in serum. Also because of dilution of the peptide. How do the authors envision this?

Are the same IP injection sites used for the peptide and later the bacteria. Is it possible that locally remaining R7 peptide is killing later applied bacteria (logically leading to all other in vivo outcomes cause less viable bacteria are present from the beginning).

It is very hard to envision how a pretreatment of R7 would work clinically. The peptide should be tested in a treatment set up where infection occurs before R7 injection.

An alternative explanation is that R7 can have immunomodulatory activity thereby boosting the immune response and protecting towards incoming bacteria. This would explain that why the peptide works without is detectably present after 6h.

-the data shows that R7 is mainly cleaved in LBP14RKR and R-L7, instead of the hypothesized LBP14RKRR and L7. This is only acknowledged by the authors in the discussion, but.... this means that all comparisons with the parent compounds LBPRKRR and L7 are relatively useless. If R-L7 has increased antibacterial activity compared to L7 (and is formed in vivo) then this could explain a lot of the in vivo data. R-L7 activity (and to lower extend LBP14RKR should be tested to a minimum in all in vivo assays).

minor point

the text states that stability is tested in mouse AND human serum but mouse and bovine serum is shown in the figure (and Suppl figure) please correct this.

Reviewer #5 (Remarks to the Author):

I have read the revised version of the manuscript in which the authors designed dual-functional peptides by linking LBP14 to L7 via different cleavable/noncleavable linkers and evaluated their antibacterial activity and LPS-neutralising ability in vitro and in a mouse infection model.

The text has been thoroughly revised by the authors and most of the questions raised by the reviewers have been adequately addressed (reviewer 1, question 1 (in part) and reviewer 2, questions 2, 3, 4, and 5).

Nevertheless, some important concerns remain (reviewer 1, question 2 and reviewer 2 question 1) and new ones should be addressed.

1) The manuscript contains incorrect premises: LPB14 was presented as an LPS-binding peptide with low antibacterial activity and L7 as a killing peptide, so it seemed advantageous to combine the two parts to obtain a peptide with both properties

Unfortunately, in a previous work (Wang, Z. et al. *Communications Biology* 3, 1-15, 2020). by the same authors, LPB14 was indicated to have a MIC of 4-8 μM against *E. coli* CVC195, whereas in this manuscript LPB14 is inactive against the same microorganism (MIC $> =64 \mu\text{M}$). Moreover, L7, the peptide presented here as the "killing domain", is almost completely inactive alone, except against *E. coli*, and less active than LPB14+RRKR. Moreover, all the parent peptides, including L7, and polymyxin B (PMB), showed stronger interaction with LPS than LPB14, weakening its definition as an LPS-binding domain (see Figure 4).

The authors should revise these definitions, which are in conflict with the facts. On the other hand without these premises the aim of this study is weakened.

2) As underlined by reviewer 1 (question 2): "The authors repeat in several places in the manuscript that the cleavable linker is responsible for the improved properties of the conjugated peptide, but the experimental data are not convincingly supporting the advantage of the cleavable linker."

In my opinion this main problem remains because the authors' answer is not convincing (Response 2: Actually, cleavable chimeric peptide R7 in vivo remained intact within a very short time. As time increases, R7 was partially cleaved into LPS-targeting and bactericidal fragments (LBP14-RKRR and L7), and formed the mixture of R7 and "LBP14RKRR+L7" simultaneously. Finally, this mixture was gradually dominated by "LBP14-RKRR +L7" due to sufficient cleavage of R7.)

The improved properties of the conjugated peptide may be due to several reasons. It could be because the peptide R7 (LPB14+RKRR+L7) is more positively charged (+14) than A7 (+8 and not +10 as indicated in Table 1) and G7 (+10) and than all other peptides. R7 is also expected to have higher activity than LPB14-RKRR + L7 simply because it is longer (35 amino acids) than the other two peptides (18 and 17 amino acids). It is widely accepted that longer AMPs, which have a high number of positive charges, are more active than short peptides (see

<https://doi.org/10.1021/acs.biochem.6b01071>; or <https://doi.org/10.1007/s13238-010-0004-3>

It is more difficult to explain why R7 remains active in vivo. Among the released products of R7 after incubation in mouse serum, by MALDI-TOF analysis, a signal of the L7 product peak was found but not the signal of the released LBP14-RKRR. L7 also rapidly degraded. Thus, there is no evidence for the presence of active peptides generated by R7 in serum. The observation of in vivo activity of R7 and L7 is quite surprising because the mice were inoculated with *E. coli* CVCC195 only 6 hours after administration of the peptides, a time during which chimeric R7 is almost completely degraded.

3) As underlined by reviewer 2 (question 1): " Peptide stability (half-life) is most important when evaluating for in vivo efficacy. The authors find that after injecting R7 peptide via IP, IV or SQ there was no peptide detected in blood of mice. The authors mention several reasons for this finding, but this is expected result from a non-stabilized peptide"

The authors responded by suggesting strategies in the text to improve stability in the future, but they

did not address the fact that the peptide was not detected in blood

The revised version omits this part (??), but the low stability was also demonstrated in in vitro tests in the presence of mouse and bovine serum. Both R7 and L7 are almost completely degraded in serum within 2 hours (80%, Figure 3b), although surprisingly, no decrease in antibacterial activity was observed during this period (Figure 3c). Since other antibacterial compounds (complement, other AMPs, lactoferrin) are present in the serum and may have contributed to the antibacterial activity, a negative control with serum should be added. The authors responded by suggesting strategies in the text to improve stability in the future, but they did not address the fact that the peptide was not detected in blood. The revised version omits this part (??), but the low stability was also demonstrated in in vitro tests in the presence of mouse and bovine serum. Both R7 and L7 are almost completely degraded in serum within 2 hours (80%, Figure 3b), although surprisingly, no decrease in antibacterial activity was observed during this period (Figure 3c). Since other antibacterial compounds (complement, other AMPs, lactoferrin) are present in the serum and may have contributed to the antibacterial activity, a negative control with serum should be added.

4) To reviewer 1, question 1: Since most of the text has been changed, I could not check whether the language has been improved, yet the manuscript still needs linguistic revision.

5) The paper remains sometimes slightly hard to read.

6) Data on the bacterial strains, particularly on E. coli CVCC195 (except the name) and on its resistances are missing.

Other minor points:

1) The amount of furin was not given (4 μ L ? New England Biolabs), and on page 11, line 158 the molar ratio between R7 and furin should be indicated.

2) When parent peptides (LBP14-RKRR and L7) were tested in parallel with R7, it is unclear whether the micromolar concentration of each of these peptides was equal to that of R7 or whether the sum of the concentrations of the two peptides was equal to that of R7. This is an important issue for the proper comparison of their activities

3) Page 7 Table 1: the charge of A7 is +8 and not +10 as incorrectly reported in the table (2 E(-1) are present)

4) Page 10 line 136: the sentence "the survival of L7 and R7 towards RAW 264.7 cells was...." should be rewritten

6) Page 24 lines 417-420. Results corresponding to these comments are not included in this version of the manuscript

7) Page 31. is polymycin B (PMB) actually polymyxin B?

8) Figure S2 (page 13) Figure S2 contains the percentages of secondary structures and not the results of stability in human serum

Referee expertise:

Referee #3: original reviewer

Referee #4: new reviewer, antimicrobial peptides

Referee #5: new reviewer, antimicrobial peptides

Reviewers' comments:

Reviewer #3 (Remarks to the Author):

The manuscript has been greatly improved. I have no further additions.

Response: Thank you for your positive evaluation.

Reviewer #4 (Remarks to the Author):

The authors have revised the manuscript according to the original comments of which most have been done satisfactorily.

I have however one major concern related to the comments on stability and the general observation that it is very hard to link the in vitro studies to the in vivo outcome.

In detail:

The chimeric R7 peptide or any of its parent molecules cannot be detected in blood. this is likely due to the short half life in blood. It's attachment or uptake to blood cells or serum components. It is hard to imagine how direct antibacterial activity can then be accomplished in serum. also because of dilution of the peptide. How do the authors envision this?

Response: Thank you for your comment. Although R7 is gradually reduced to ~20% after incubated with serum in vitro at 8h (Fig. 3b), it does not affect the antibacterial activity, and it remains the same as at 0 h (Fig. 3c). This result indicated that ~80% undetectable R7 monomer still has antibacterial activity.

Fig. 3. Stability and release of peptides in mouse serum. b The remaining rate of R7 and L7 in mouse serum. c The antibacterial activity of R7 and L7 against MDR *E. coli* in mouse serum detected by an inhibition assay.

In the previous manuscript, we discussed the potential factors (low concentration, interaction with serum proteins, degradation) that prevented the detection of R7 by HPLC *in vivo*. In the revised manuscript, we employed a more sensitive Liquid Chromatograph Mass Spectrometer (HPLC-MS) to identify R7 and potential degradation products in mice. Table S4 reveals the detection of a "RMKKLAAPS" fragment from the C-terminal region of R7, providing evidence of R7 degradation in mice.

Table S4 Fragment analysis of R7 by LC-MS in mouse

Sequence	# PSMs	Modifications	q-Value	MH+ [Da]	RT [min]
RMKKLAAPS	1		0	1001.59220	11.21

Nevertheless, the possibility exists that additional active fragments may remain undetectable due to their binding affinity with serum proteins. Additionally, with the blood flowing, R7 will spread to various organs and tissues of mice to play a bacteriostatic role, which may also lead to the decrease of its concentration in blood. To avoid the detection difficulty caused by the flow and diffusion of R7 in mice, we further used MALDI-TOF MS with higher resolution to analyze R7 degradation products in blood in this revised manuscript. Table S5 demonstrates that R7 is primarily cleaved into two dual-function fragments within the serum. The first fragment, spanning positions 1 to 15, which was similar to LBP14 that exhibits the ability to target LPS, whereas the second fragment, primarily located from positions 18 to 35, which was similar to L7 that displays the antibacterial activity.

Table S5 Fragment analysis of R7 by MALDI-MS in mouse ^a.

Original peptide	Fragment mass (Da)	Assignment
R7	4477.602	RVQGRWKVRSFFFKR [↑] RFFKAWRWAWRMKKLAAPS
	3310.804	ASFFFKR [↑] RFFKAWRWAWRMKKLAAPS
	2289.3670	RFKAWRWAWRMKKLAAPS [↑]

1858.0977		AWRWAWRMKKLAAPS
1637.0069	RVQGRWKVRASFF	
1480.9025	↑VQGRWKVRASFF ^b	
	↑RWKVRASFFKR ^b	
1184.7919	RVQGRWKVR↑	
929.6156	RVQGRWK	

^a Peptides with observed intermediate fragments are listed here.

^b These fragments share same molar mass.

The arrow indicates the location where the cleavable sites occurred at R.

We also use "ex vivo" model to simulate the "in vivo" environment, as shown in Table S7. In general, R7 is indeed degraded into two parts after incubate with serum, one is the N-terminal bactericidal fragment, and the other is the C-terminal binding LPS fragment.

Table S7 top five degradation combined fragments analysis of R7 in serum

R7	RVQGRWKVRASFFK-RKRR- FKAWRWAWRMKKLAAPS	
1	RVQGRWKVRASFFK-RKR	R- FKAWRWAWRMKKLAAPS
2	RVQGRWKVRASFF	K-RKRR- FKAWRWAWRMKKLAAPS
3	RVQGRWKVRASFFK-RKRR- FK	AWRWAWRMKKLAAPS
4	RVQGRWKVRASFFK-R	KRR- FKAWRWAWRMKKLAAPS
5	RVQGRWKVRASFFK-RK	RR- FKAWRWAWRMKKLAAPS

^a Table S7 is summarized from table S3.

Are the same IP injection sites used for the peptide and later the bacteria. Is it possible that locally remaining R7 peptide is killing later applied bacteria (logically leading to all other in vivo outcomes cause less viable bacteria are present from the beginning.

Response: No, in fact, to avoid false positive results due to the same injection sites, we refer to the previous method of intraperitoneal injection: If animals are to receive repeated intraperitoneal (IP) injections, alternate the site of injection (Machholz et al. 2012). We injected R7 in the lower right quadrant and the bacteria in the lower left quadrant after 6 hours. In addition, in practice, an extra step is added at the end of the intraperitoneal injection: gently kneading the abdominal site with the fingers to allow the drug to be transferred and absorbed as soon as possible with the peristalsis of the organs.

Quadrants of the ventral abdomen.

Page 35, Line 604-607: “Briefly, the mice (five mice/group) were intraperitoneally injected with 7 $\mu\text{mol/kg}$ peptides (R7, L7, LBP14-RKRR, LBP14-RKRR+L7) and challenged with a lethal dose of MDR *E. coli* CVCC195 (0.5×10^9 CFU/mouse) or LPS (13.5 mg/kg) after 6 h peptides administration.”
was changed to

“As the previous method of intraperitoneal injection⁶⁰, The left abdominal cavity of mice (five mice/group) was injected with 7 $\mu\text{mol/kg}$ (minimum effect dose in preliminary test (Fig. S3)) chemic peptide-R7, L7, LBP14-RKRR, the “LBP14-RKRR+L7” combination or PBS, followed by the right intraperitoneal injection with a lethal dose of MDR *E. coli* CVCC195 or LPS (13.5 mg/kg) after 6 h

It is very hard to envision how a pretreatment of R7 would work clinically. The peptide should be tested in a treatment set up where infection occurs before R7 injection.

Response: Thank you for your comment. We have added the result of R7 treatment experiment as supplementary material Fig. S4 in the new revised version. In initial mouse experiments, we used R7 as a therapeutic agent against *E. coli* and LPS infected mouse, but R7 was almost ineffective in treatment groups. For *E. coli* infected mice, all mice in the R7 treated group died within 5 days (Fig. S4a); similarly, for LPS infection, the survival rate in the R7 treatment group was only 20% (Fig. S4b). According to a previous report, lactoferrin was effective when administered 24 h before infection (Palumbo et al. 2010). Therefore, we changed our thinking and applied R7 to prevention experiments. The results showed that R7 prevention group did exert an in vivo inhibitory effect on *E. coli* and LPS, and the survival rate of mice was 100% (Fig. 5 and Fig. 7). In our laboratory, some bovine lactoferrin has been used as a feed additive, which enhances the growth performance, reduced diarrhea rate induced by bacteria (Ma et al. 2023). Therefore, we speculate that R7 may also serve as a beneficial feed additive in the future. For example, R7 could be added to piglet feed during weaning to prevent *E. coli* infections and neutralize the LPS toxicity.

Fig. S4. Therapeutic efficacy of cleavable chimeric peptide R7 in the mice challenged with MDR *E. coli* or LPS After the intraperitoneal injection with MDR *E. coli* or LPS at a concentration of LD₁₀₀, the mice were treated with 7

$\mu\text{mol/kg}$ R7 at 0.5 h and 8 h, respectively. The mouse survival was recorded for 7 d. The mice injected with MDR *E. coli*/LPS or saline were used as negative or blank controls, respectively

Fig. 5. Efficacy of cleavable chimeric peptide R7 in the mice challenged with MDR *E. coli*. a Survival of mice. Mice were intraperitoneally injected with 7 $\mu\text{mol/kg}$ different peptides (including R7, L7, LBP14-RKRR, the “LBP14-RKRR+L7” combination) at 0 h (five mice/group), followed by injection with MDR *E. coli* CVCC195 (0.5×10^9 CFU/mouse) at 6 h. Survival was recorded for 7 d. Two biological replicates were done.

Fig. 7. Efficacy of cleavable chimeric peptide R7 in the mice challenged with LPS. a Survival of mice. Mice were intraperitoneally injected with different peptides (7 $\mu\text{mol/kg}$) at 0 h (five mice/group), followed by injection with LPS (13.5 mg/kg of body weight) at 6 h. Survival was recorded for 7 d. Two biological replicates were done

Page 17, Line 236-238: “Our previously constructed mouse intraperitoneal infection model was used to investigate the in vivo antimicrobial activity of peptides¹³. The mice were intraperitoneally injected with 7 $\mu\text{mol/kg}$ (minimum effect dose in preliminary test (Fig. S3)) chemic peptide-R7, L7, LBP14-RKRR, the “LBP14-RKRR+L7” combination or PBS, followed by an intraperitoneal injection with a lethal dose of MDR *E. coli* CVCC195 after 6 h.”

was changed to

“Our previously constructed mouse intraperitoneal infection model was used to investigate the in vivo antimicrobial activity of peptides¹³. We first used R7 as a therapeutic agent against *E. coli* and LPS infected mouse, and the results showed that R7 was almost ineffective ($\leq 20\%$) in treatment groups (Fig. S3). Therefore, we changed our thinking and applied R7 to prevention experiments. The mice were intraperitoneally injected with 7 $\mu\text{mol/kg}$ (minimum effective dose in preliminary test (Fig. S4)) chemic peptide-R7, L7, LBP14-RKRR, the “LBP14-RKRR+L7” combination or PBS,”

An alternative explanation is that R7 can have immunomodulatory activity thereby boosting the immune response and protecting towards incoming bacteria. This would explain that why the peptide works without is detectably present after 6h.

Response: Thank you for your comment. R7 is a peptide derived from lactoferrin. In previous reports, one of the important functions of lactoferrin is immunomodulation. To verify your suggestion, we detected some key proteins related to inflammation pathway through western blot assay. As shown in Fig 3, the results show that the proteins related to NF- κ B pathways have changed significantly. The R7 more promoted LPS-induced I κ B α levels and reduced NF- κ B p65 phosphorylation than L7 alone (Fig. S5a, b, c). Additionally, neither R7 nor L7 have effect on the Erk1/2 pathways (Fig. S5a, d). These results demonstrate that R7 exert their anti-inflammatory effects through selective inhibition of the NF- κ B inflammatory signaling pathways.

Fig. S5 Effects of R7 and L7 on LPS-induced NF- κ B and MAPK signaling pathways in lung tissues. The mice were pretreated with R7 and L7, and followed by injected with LPS. a The protein levels of p65, p-p65, ERK1/2, p-ERK1/2, and I κ B α in lungs were analyzed by western blotting. b Densitometric analysis of p-ERK1/2/ERK1/2 ratio. c. Densitometric analysis of p-p65/p65 ratio. d Densitometric analysis of I κ B α / β -actin ratio. All data were analyzed with one-way ANOVA, and data are means \pm SD (n = 3). *p*-values < 0.05 were considered significant. **, *p* < 0.01, ***, *p* < 0.001, ****, *p* < 0.0001.

Page 25, Line 387: “Furthermore, the effects of R7 and L7 on LPS-induced signal pathways involved

in inflammation were detected by Western blotting (Fig. S5). Compared to L7, R7 significantly promoted (27.2%) LPS-induced I κ B α levels and inhibited (47.0%) the LPS-induced translocation of the NF- κ B p-p65 subunit from the cytoplasm to the nucleus (Fig. S5 a, b, c). However, ERK1/2 phosphorylation was not markedly attenuated by R7 and L7 (Fig. S5 a, d), suggesting that R7 and L7 didn't regulate MAPK pathway. These results demonstrate that R7 exert the anti-inflammatory effects through the selective inhibition of the NF- κ B inflammatory signaling pathways." **was added**

Page 290, Line 470: "chimeric peptide R7 with the cleavable linker of "RKRR" is one of potential endotoxemia therapeutics, which may be attributed to two factors: i) the strong ability of LBP14 in R7 to neutralize LPS (Fig. 4a); R7 may be cleaved by cellular furin in mice, followed by release of active LBP14-RKRR and L7, thereby more potently suppressing the LPS toxicity and promoting the mouse survival than L7 alone^{13,52,53}; ii) higher IAP activity induced by R7 than LBP14-RKRR and L7 (Fig. 5g, Fig. 7f), which can dephosphorylate and detoxify LPS by binding to LPS (Fig. 4) and directly altering the molecular structure of LPS, leading to the amelioration of inflammatory activity of LPS^{13,16}." **was changed to** "chimeric peptide R7 with the cleavable linker of "RKRR" is one of potential endotoxemia therapeutics, which may be attributed to three factors: i) the strong ability of LBP14 in R7 to neutralize LPS (Fig. 4a); R7 may be cleaved by cellular furin in mice, followed by release of active LBP14-RKRR and L7, thereby more potently suppressing the LPS toxicity and promoting the mouse survival than L7 alone^{13,52,53}; ii) higher IAP activity induced by R7 than LBP14-RKRR and L7 (Fig. 5g, Fig. 7f), which can dephosphorylate and detoxify LPS by binding to LPS (Fig. 4) and directly altering the molecular structure of LPS, leading to the amelioration of inflammatory activity of LPS^{13,16}."; iii) The R7 more promoted LPS-induced I κ B α levels and reduced NF- κ B p65 phosphorylation than L7 (Fig. S5)."

Page 36, Line 622: Western blot assay description is added in Method section as follows:

Additionally, the lung tissues of 24 h were homogenized and suspended in 5 M HEPES buffer, and the protein levels of supernatants were quantified using a Bradford protein assay kit. A total of 30 μ g proteins were analyzed on a sodium dodecyl sulfate-polyacrylamide gel electrophoresis gel and were immunoblotted onto PVDF membranes, followed by an incubation with the primary antibodies of p65, p-p65, extracellular signal-regulated kinase (ERK)1/2, p-ERK1/2 and I κ B α overnight at 4°C. After washing, the blots were incubated with the peroxidase conjugated goat-rabbit or goat-mouse antibodies. Relative protein expression levels were quantified by a densitometric measurement of chemiluminescence (ECL) reaction band. β -actin was used as the internal reference.

the data shows that R7 is mainly cleaved in LBP14RKRR and R-L7, instead of the hypothesized LBP14RKRR and L7. This is only acknowledged by the authors in the discussion, but... this means

that all comparisons with the parent compounds LBPRKRR and L7 are relatively useless. If R-L7 has increased antibacterial activity compared to L7 (and is formed in vivo) then this could explain a lot of the in vivo data. R-L7 activity (and to lower extend LBP14RKR should be tested to a minimum in all in vivo assays.

Response: We understand your concern about the different antibacterial activities of R-L7 and L7. As your suggestion, we synthesized LBP14-RKR and R-L7 peptides, respectively, and compared their antibacterial activity with LBP14-RKRR and L7. The results showed that the antibacterial activities of LBP14-RKR and R-L7 against the tested strains were consistent with those of LBP14-RKRR and L7, respectively (Table S6). In addition, we identified the top five combination fragments resulting from degradation (Table S7). In fact, except for product pair LBP14-RKR and R-L7(1), the effect of “2-5” list in Table S7 also need to be evaluated in vivo and in vitro, and we plan to conduct this research in the future due to the huge workload.

Table S6 the MIC (μM) values of LBP14-RKRR, LBP14-RKR, L7 and R-L7.

Strains	LBP14-RKRR	LBP14-RKR	L7	R-L7
Escherichia coli CVCC195	2.26	2.26	2.26	2.26
S. typhimurium CVCC533	4.53	4.53	18.1	18.1
Staphylococcus aureus ATCC43300	4.53	4.53	18.1	18.1
S. aureus CVCC546	9.06	9.06	18.1	18.1

Table S7 top five degradation combined fragments analysis of R7 in serum

R7	RVQGRWKVRASFFK-RKRR- FKAWRWAWRMKKLAAPS	
1	RVQGRWKVRASFFK-RKR	R- FKAWRWAWRMKKLAAPS
2	RVQGRWKVRASFF	K-RKRR- FKAWRWAWRMKKLAAPS
3	RVQGRWKVRASFFK-RKRR- FK	AWRWAWRMKKLAAPS
4	RVQGRWKVRASFFK-R	KRR- FKAWRWAWRMKKLAAPS
5	RVQGRWKVRASFFK-RK	RR- FKAWRWAWRMKKLAAPS

Minor point: the text states that stability is tested in mouse AND human serum but mouse and bovine serum is shown in the figure (and Suppl figure) please correct this.

Response: Thanks for your reminder. It was a mistake. “bovine serum” in the supplementary material Fig. S2 has been replaced by “human serum”.

Reviewer #5 (Remarks to the Author):

I have read the revised version of the manuscript in which the authors designed dual-functional peptides by linking LBP14 to L7 via different cleavable/noncleavable linkers and evaluated their antibacterial activity and LPS-neutralising ability in vitro and in a mouse infection model.

The text has been thoroughly revised by the authors and most of the questions raised by the reviewers have been adequately addressed (reviewer 1, question 1 (in part) and reviewer 2, questions 2, 3, 4, and 5).

Nevertheless, some important concerns remain (reviewer 1, question 2 and reviewer 2 question 1) and new ones should be addressed.

Response: Thank you. As suggested, we answered these two comments carefully again as follow:

Reviewer 1, Question 2: The main conclusion repeated several places in the manuscript that the cleavable linker is responsible for the improved properties on the conjugated peptide appears not very likely - all data indicate that the beneficial activities arise from the combination of these moieties into a more stable and more active peptide. Thus R7 is more potent than L7 and LBP-14 with respect to both antibacterial and LPS-binding properties (Fig. 5A), and hence the experimental data are not convincingly supporting the advantage of the cleavable linker.

Response: Thank you for your comment. A previous study demonstrated that P14KanS with a cleavable linker was more potent than the 1:1 mixture of parental molecule P14LRR and kanamycin, which was used to inhibit *S. enteritidis* (Brezden, et al., 2016). Similarly, in our work, R7 (which will be cleaved into “LBP14RKRR+L7” forms in vivo) showed higher potency than the cocktail (a 1:1 mixture of LBP14-RKRR and L7). This indicates that the presence of a cleavable linker is crucial for the activity of R7. To further evaluate the importance of the cleavable linker for R7’ activity, we designed two non-cleavable chimeric peptides-A7 and G7 as controls by connecting LBP14 (LPS-targeting domain) with L7 (killing domain) by the non-cleavable flexible “G4S” and rigid “(EA3K)2” linkers, respectively. The “G4S” linker allows for flexible mobility of the two functional domains LBP14 and L7 while the “(EA3K)2” linker can maintain rigid spatial distance between LBP14 and L7. As shown in Table 2, the non-cleavable chimeric peptides-A7 and G7 exhibited weak antibacterial activity ($\geq 18.1 \mu\text{M}$). This suggests that the cleavable linker (“RKRR”) is more suitable for the chimerization of L7 with LBP14 than the non-cleavable linkers, including “(EA3K)2” and “G4S”.

Reviewer 2 Question 1 Peptide stability (half-life) is most important when evaluating for in vivo efficacy. The authors find that after injecting R7 peptide via IP, IV or SQ there was no peptide detected in blood of mice. The authors mention several reasons for this finding, but this is expected result from a non-stabilized peptide. Most L-peptides will have a half-life of 15 minutes or even less when

injected. This is the main issue with this study, and authors should comment on it. Some strategies to stabilize the peptide in the future could be proposed.

Response: Yes, you are right. In the new revised manuscript, we used more accurate mass spectrometry (LC-MS and MALDI-TOF MS) to detect the metabolites of R7 in mice, and added some comment regarding peptide stability and strategies to stabilize the peptide in the Discussion section as follow:

Pages 27, Line 426: “Furthermore, we used more sensitive liquid chromatograph mass spectrometer (LC-MS) to detect R7 and its possible degradation products in mice. As shown in Table S4, a fragment “RMKKLAAPS” from the C-terminal of R7 was detected, which indicated that R7 was indeed degraded in mice. We further used MALDI-TOF MS with higher resolution to analyze R7 degradation products in blood. As shown in Table S5, R7 was cleaved into two dual-function fragments in the serum principally, including one from position 1 to 15 that can target LPS and the other mainly from position 18 to 35 that has antibacterial activity. Notably, 6 out of 8 Arg residues in R7 especially prone to protease cleavage in mice (Table S5), which should be replaced with non-proteolytic amino acids such as ornithine and D-Arg to stabilize peptides in blood in the next study.”

Table S4 Fragment analysis of R7 by LC-MS in mouse

Sequence	# PSMs	Modifications	q-Value	MH+ [Da]	RT [min]
RMKKLAAPS	1		0	1001.59220	11.21

Table S5 Fragment analysis of R7 by MALDI-MS in mouse ^a.

Peptide	Fragment mass (Da)	Assignment
R7	4477.602	RVQGRWKVRASFFKRRFKAWRWAWRMKKLAAPS
	3310.804	ASFFKRRFKAWRWAWRMKKLAAPS
	2289.3670	RFKAWRWAWRMKKLAAPS
	1858.0977	AWRWAWRMKKLAAPS
	1637.0069	RVQGRWKVRASFF
	1480.9025	VQGRWKVRASFF ^b
		RWKVRASFFKR ^b
	1184.7919	RVQGRWKVR
	929.6156	RVQGRWK

^a Peptides with observed intermediate fragments are listed here.

^b These fragments share same molar mass.

The arrow indicates the location where the cleavable sites occurred at R

Pages 27-28, Line 428-440: “In fact, several strategies have been proposed to improve the stability of AMPs. Substituting some enzyme-sensitive L-amino acids with D-enantiomers is the simplest way to improve the protease stability of AMPs. In previous studies, L-amino acids in AMPs such as WLBU2, W3R, and N6NH2 were substituted with D-enantiomers (Di et al., 2020; Li et al., 2019; Li, et al., 2020), and it was found that these peptides with D-amino acids had higher stability in trypsin, simulated gastric/intestinal fluid, proteinase K, and mouse serum than the parent peptides. In addition, rational arrangement of amino acids is also an effective strategy. For example, Pro containing a pyrrole ring is resistant to trypsin and chymotrypsin by steric hindrance; therefore, Pro is usually placed after amino acids sensitive to protease for better stability (Zhu et al., 2020).” Recently, nanotechnology has been also used to improve the protease and serum stability of AMPs. Zhou et al. designed two self-assembled peptide EN (nanofibrils structure) and NE (micelle structure), and found that self-assembly can effectively protect EN and NE from the degradation of proteases in serum (Zhou et al., 2017).

The manuscript contains incorrect premises: LPB14 was presented as an LPS-binding peptide with low antibacterial activity and L7 as a killing peptide, so it seemed advantageous to combine the two parts to obtain a peptide with both properties

Unfortunately, in a previous work (Wang, Z. et al. Communications Biology 3, 1-15, 2020). by the same authors, LPB14 was indicated to have a MIC of 4-8 μ M against E. coli CVC195, whereas in this manuscript LPB14 is inactive against the same microorganism (MIC \geq 64 μ M). Moreover, L7, the peptide presented here as the "killing domain", is almost completely inactive alone, except against E. coli, and less active than LPB14+RRKR. Moreover, all the parent peptides, including L7, and polymyxin B (PMB), showed stronger interaction with LPS than LPB14, weakening its definition as an LPS-binding domain (see Figure 4).

The authors should revise these definitions, which are in conflict with the facts. On the other hand without these premises the aim of this study is weakened.

Response: Thank you for your valuable comment. In our previous articles (Wang, Z. et al. Communications Biology 3, 1-15, 2020), the MIC values of LBP14 for *E. coli* CVCC195, *S. typhimurium* CVCC533 and *Staphylococcus aureus* ATCC43300 were 8 μ g/ml (4.53 μ M), 32 μ g/ml (18.1 μ M) and 64 μ g/ml (36.2 μ M). Conversely, in the current manuscript, the MIC values of LBP14 are 64 μ M, $>$ 64 μ M and $>$ 64 μ M, respectively. This error is caused by the different measurement methods of peptide concentration. In our previous work (Wang, Z. et al. Communications Biology 3, 1-15, 2020), we used Bradford method to measure the peptide concentration, but in this work, we used nanodrop to measure the concentration. That is why the MIC values are different. We have corrected all peptide concentration (by Bradford method), and re-measured the MIC and MBC values in Table 2. L7 exhibited bactericidal and LPS-binding effects (Table 2, Fig.4B), but it was not flawless. To enhance its

efficacy, we incorporated LBP14. While L7 itself can bind to LPS, the inclusion of LBP14 serves as a targeted carrier to bacterial membranes, which is advantageous. Upon linking LBP14 and L7, R7 demonstrated an increased capacity to bind to LPS, and the presence of LBP14 provided additional LPS binding sites for R7 (Fig. 4B).

Table 2 MIC and MBC values of the peptides.

	LBP14		LBP14-RKRR		L7		R7		LBP14 + L7 ^a		LBP14-RKRR + L7 ^a		PMB		A7 ^b		G7 ^c	
	MIC (μ M)	MBC (μ M)	MIC (μ M)	MBC (μ M)	MIC (μ M)	MBC (μ M)	MIC (μ M)	MBC (μ M)	MIC (μ M)	MBC (μ M)	MIC (μ M)	MBC (μ M)	MIC (μ M)	MBC (μ M)	MIC (μ M)	MBC (μ M)	MIC (μ M)	MBC (μ M)
Gram-negative bacteria																		
Escherichia coli CVCC195 ^d	4.53	4.53	2.26	2.26	2.26	4.53	1.13	1.13	1.13	2.26	1.13	2.26	1	1	36.2	36.2	18.1	18.1
S. typhimurium CVCC533	18.1	18.1	4.53	128	18.1	18.1	1.13	2.26	4.53	4.53	4.53	4.53	8	8	>36.2	>36.2	>36.2	>128
Gram-positive bacteria																		
Staphylococcus aureus ATCC43300 ^b	36.2	36.2	4.53	64	18.1	18.1	2.26	2.26	9.06	9.06	2.26	2.26	>64	>64	>36.2	>36.2	>36.2	>36.2
S. aureus CVCC546 ^b	36.2	>36.2	9.06	9.06	18.1	36.2	2.26	2.26	4.53	4.53	2.26	2.26	>64	>64	>36.2	>36.2	>36.2	>36.2

^a 1:1 mixture of LBP14 (or LBP14-RKRR) and L7. ^b A7: connect LBP14 and L7 via rigid linker (EA₃K)₂⁷. ^c G7: connect LBP14 and L7 via flexible linker G₄S₇⁷. ^d The strains are resistant to different antibiotics^{7,33}. The MDR *E. coli* CVCC195 strain is resistant to doxycycline, benzocillin, penicillin, lincomycin, clindamycin, vancomycin, chloramphenicol, erythromycin, amikacin, streptomycin, gentamicin and sulfamethoxazole, respectively. The *S. aureus* ATCC43300 strain is resistant to lincomycin, amoxicillin, amikacin, ampicillin, oxacillin, erythrocine, sulfisoxazole, neomycin, azithromycin, kanamycin, gentamicin, and penicillin, respectively. The *S. aureus* CVCC546 strain is resistant to tetracycline, bacitracin, and sulfisoxazole, respectively.

Fig. 4. Interaction between peptides and LPS. (b) Putative binding sites of LBP14-RKRR, L7, and R7 with LPS. Docking was performed using Autodock vina. Up: complex structures of LBP14-RKRR, L7 or R7 and LPS. Oxygen, carbon, hydrogen atoms are indicated as red, green and white, respectively. Down: H-bonding between the peptides and LPS. LPS and peptide chains are shown as green and blue, respectively. The hydrogen bonds are shown in yellow dotted lines.

As underlined by reviewer 1 (question 2): "The authors repeat in several places in the manuscript that the cleavable linker is responsible for the improved properties of the conjugated peptide, but the experimental data are not convincingly supporting the advantage of the cleavable linker."

In my opinion this main problem remains because the authors' answer is not convincing (Response 2: Actually, cleavable chimeric peptide R7 in vivo remained intact within a very short time. As time increases, R7 was partially cleaved into LPS-targeting and bactericidal fragments (LBP14-RKRR and L7), and formed the mixture of R7 and "LBP14RKRR+L7" simultaneously. Finally, this mixture was gradually dominated by "LBP14-RKRR +L7" due to sufficiently cleavage of R7.)

Response: Thanks for your comment and sorry for the previous unsatisfactory response, we answered this question carefully again. A previous study demonstrated that P14KanS with a cleavable linker was

more potent than the 1:1 mixture of parental molecule P14LRR and kanamycin, which was used to inhibit *S. enteritidis* (Brezden, et al., 2016). Similarly, in our work, R7 (which will be cleaved into “LBP14RKRR+L7” forms in vivo) showed higher potency than the cocktail (a 1:1 mixture of LBP14-RKRR and L7). When the 1: 1 mixture of LBP14-RKRR+L7 was directly used to treat mice, the survival rate was 40~80%, less than R7(100%) (Fig 5a, Fig.7a). This indicates that the presence of a cleavable linker and its cleavage in vivo is crucial for the activity of R7. To further evaluate the importance of the cleavable linker for R7’ activity, we designed two non-cleavable chimeric peptides-A7 and G7 as controls by connecting LBP14 (LPS-targeting domain) with L7 (killing domain) by the non-cleavable flexible “G4S” and rigid “(EA3K)2” linkers, respectively. The “G4S” linker allows for flexible mobility of the two functional domains LBP14 and L7 while the “(EA3K)2” linker can maintain rigid spatial distance between LBP14 and L7. As shown in Table 2, the non-cleavable chimeric peptides-A7 and G7 exhibited weak antibacterial activity ($\geq 18.1 \mu\text{M}$). This suggests that the cleavable linker (“RKRR”) is more suitable for the chimerization of L7 with LBP14 than the non-cleavable linkers, including “(EA3K)2” and “G4S”.

Fig. 5. Efficacy of cleavable chimeric peptide R7 in the mice challenged with MDR *E. coli*. a Survival of mice. Mice were intraperitoneally injected with 7 $\mu\text{mol/kg}$ different peptides (including R7, L7, LBP14-RKRR, the “LBP14-RKRR+L7 combination) at 0 h (five mice/group), followed by injection with MDR *E. coli* CVCC195 (0.5×10^9 CFU/mouse) at 6 h. Survival was recorded for 7 d. Two biological replicates were done.

Fig. 7. Efficacy of cleavable chimeric peptide R7 in the mice challenged with LPS. a Survival of mice. Mice were intraperitoneally injected with different peptides (7 $\mu\text{mol/kg}$) at 0 h (five mice/group), followed by injection with LPS (13.5 mg/kg of body weight) at 6 h. Survival was recorded for 7 d. Two biological replicates were done

The improved properties of the conjugated peptide may be due to several reasons. It could be because

the peptide R7 (LPB14+RKRR+L7) is more positively charged (+14) than A7 (+8 and not +10 as indicated in Table 1) and G7 (+10) and than all other peptides. R7 is also expected to have higher activity than LPB14-RKRR + L7 simply because it is longer (35 amino acids) than the other two peptides (18 and 17 amino acids). It is widely accepted that longer AMPs, which have a high number of positive charges, are more active than short peptides

(see <https://doi.org/10.1021/acs.biochem.6b01071>; or <https://doi.org/10.1007/s13238-010-0004-3>)

Response: Thank you for your comment. We systematically compared the effects of two types of linkers (cleavable and non-cleavable) on the activity of chimerization of L7 with LBP14, and found that R7 with cleavable linker showed the highest activity against tested *E. coli*, *S. typhimurium* and *S. aureus*. compared with A7/G7 with non-cleavable linkers. In addition, the in vivo effect of R7 is better than that of directly injecting the cocktail of LBP14+L7 mixed in vitro in advance (Fig. 5a, Fig. 7a). This results indicated that the presence of a cleavable linker is crucial for the activity of R7. Moreover, we agree with your views that more charges will increase the activity of peptides. We added relevant descriptions in Result section. However, the influence of length on antibacterial activity is not applicable in the current work. Although the length of A7 and G7 is greater than R7 and L7, the antibacterial activity is far less than R7 and L7 (Table 2).

Page 26, Line 339-402: “compared to parental peptides LBP14-RKRR and L7, chimeric peptide R7 with a cleavable linker (“RKRR”) displayed more potent antibacterial activity ($\geq 2\sim 4$ -fold increase) against the tested *E. coli*, *S. typhimurium*, and *S. aureus* (Table 2)” **was changed to** “compared to parental peptides LBP14-RKRR (+9) and L7 (+5), chimeric peptide R7 (+14) with a cleavable linker (“RKRR”) displayed more potent antibacterial activity ($\geq 2\sim 4$ -fold increase) against the tested *E. coli*, *S. typhimurium*, and *S. aureus* (Table 2), this may be attributed to the fact that R7 with more positive charges has more potent interaction with bacterial membranes;”

It is more difficult to explain why R7 remains active in vivo. Among the released products of R7 after incubation in mouse serum, by MALDI-TOF analysis, a signal of the L7 product peak was found but not the signal of the released LBP14-RKRR. L7 also id rapidly degraded. Thus, there is no evidence for the presence of active peptides generated by R7 in serum. The observation of in vivo activity of R7 and L7 is quite surprising because the mice were inoculated with E. coli CVCC195 only 6 hours after administration of the peptides, a time during which chimeric R7 is almost completely degraded.

As underlined by reviewer 2 (question 1): “Peptide stability (half-life) is most important when evaluating for in vivo efficacy. The authors find that after injecting R7 peptide via IP, IV or SQ there was no peptide detected in blood of mice. The authors mention several reasons for this finding, but this is expected result from a non-stabilized peptide”

The authors responded by suggesting strategies in the text to improve stability in the future, but they

did not address the fact that the peptide was not detected in blood

The revised version omits this part (??), but the low stability was also demonstrated in in vitro tests in the presence of mouse and bovine serum.

Response: Thank you for your comment. As shown in Fig. 3b, although R7 is gradually reduced to ~20% after incubated with serum in vitro at 8h, it does not affect the antibacterial activity, and it remains the same as at 0 h (Fig. 3c). This result shows that ~80% undetectable R7 monomer still has antibacterial activity.

Fig. 3. Stability and release of peptides in mouse serum. b The remaining rate of R7 and L7 in mouse serum. c The antibacterial activity of R7 and L7 against MDR E. coli in mouse serum detected by an inhibition assay.

In the previous manuscript, we discussed the potential factors (low concentration, interaction with serum proteins, degradation) that prevented the detection of R7 by HPLC in vivo. In the revised manuscript, we employed a more sensitive Liquid Chromatograph Mass Spectrometer (LC-MS) to identify R7 and potential degradation products in mice. Table S4 reveals the detection of a "RMKKLAAPS" fragment from the C-terminal region of R7, providing evidence of R7 degradation in mice.

Table S4 Fragment analysis of R7 by LC-MS in mouse

Sequence	# PSMs	Modifications	q-Value	MH+ [Da]	RT [min]
RMKKLAAPS	1		0	1001.59220	11.21

Nevertheless, the possibility exists that additional active fragments may remain undetectable due to their binding affinity with serum proteins. Additionally, with the blood flowing, R7 will spread to various organs and tissues of mice to play a bacteriostatic role, which may also lead to the decrease of its concentration in blood. To avoid the detection difficulty caused by the flow and diffusion of R7 in mice, we further used MALDI-TOF MS with higher resolution to analyze R7 degradation products in mice in this revised manuscript. Table S5 demonstrates that R7 is primarily cleaved into two dual-function fragments within the serum. The first fragment, spanning positions 1 to 15, similar to LBP14 that exhibits the ability to target LPS, whereas the second fragment, primarily located from positions 18 to 35, similar to L7 that displays the antibacterial activity.

Table S5 Fragment analysis of R7 by MALDI-MS in mouse ^a.

Peptide	Fragment mass (Da)	Assignment
R7	4477.602	RVQGRWKVRASFFKRRRFKAWRWAWRMKKLAAPS
	3310.804	ASFFKRRRFKAWRWAWRMKKLAAPS
	2289.3670	RFKAWRWAWRMKKLAAPS
	1858.0977	AWRWAWRMKKLAAPS
	1637.0069	RVQGRWKVRASFF
	1480.9025	VQGRWKVRASFF ^b
		RWKVRASFFKR ^b
	1184.7919	RVQGRWKVR
	929.6156	RVQGRWK

^a Peptides with observed intermediate fragments are listed here.

^b These fragments share same molar mass.

The arrow indicates the location where the cleavable sites occurred at R.

We also use "ex vivo" model to simulate the "in vivo" environment, as shown in Table S3. In general, R7 is indeed degraded into two parts after incubate with serum, one is the N-terminal bactericidal fragment, and the other is the C-terminal binding LPS fragment.

Table S3 top five degradation combined fragments analysis of R7 in serum

R7	RVQGRWKVRASFFK-RKRR- FKAWRWAWRMKKLAAPS	
1	RVQGRWKVRASFFK-RKR	R- FKAWRWAWRMKKLAAPS
2	RVQGRWKVRASFF	K-RKRR- FKAWRWAWRMKKLAAPS
3	RVQGRWKVRASFFK-RKRR- FK	AWRWAWRMKKLAAPS
4	RVQGRWKVRASFFK-R	KRR- FKAWRWAWRMKKLAAPS
5	RVQGRWKVRASFFK-RK	RR- FKAWRWAWRMKKLAAPS

Both R7 and L7 are almost completely degraded in serum within 2 hours (80%, Figure 3b), although surprisingly, no decrease in antibacterial activity was observed during this period (Figure 3c). Since other antibacterial compounds (complement, other AMPs, lactoferrin) are present in the serum and may have contributed to the antibacterial activity, a negative control with serum should be added.

Response: Thank you for your suggestion, in fact, we have made a negative control of the serum. The following picture is the original experimental data, the single serum cannot form a bacteriostatic circle on the MHA plate containing *E. coli* CVCC 195, which indicates that the antimicrobial active substance comes from the complete or degraded fragment of L7 or R7.

To reviewer 1, question 1: Since most of the text has been changed, I could not check whether the language has been improved, yet the manuscript still needs linguistic revision.

Response: The language revision has been done in the new version. Such as:

“Pathogenic *Escherichia coli*, a Gram-negative bacterium, leads to ~1.7 billion diarrhea diseases that occur in both humans and animals worldwide each year ¹. Worryingly, multidrug-resistant (MDR) *E. coli* was difficult to be killed by the last-resort antibiotics, polymyxins ^{2,3}. Moreover, pathogenic Gram-negative bacterial infection is always accompanied by the release of lipopolysaccharide (LPS) from the cell wall ^{4,5}. LPS, also termed endotoxin, is a major constituent of the outer membranes of Gram-negative bacteria and construct a prime natural barrier that can prevent the entry of antibacterial agents into bacteria ^{6,7}. Simultaneously, LPS is an important virulence factor pathogen-associated molecular pattern (PAMP) ^{8,9}, which usually stimulates immune cells to activate related signaling pathways and produces uncontrolled cytokine storm, leading to severe sepsis with a mortality rate of 30~50% ¹⁰⁻¹². Common antibiotics, such as ampicillin and polymyxin B, cannot effectively neutralize LPS ^{13,14} due to lack of specific binding sites and LPS modifications, and even accelerate its release from bacteria ^{15,16}. Therefore, it is very urgent to find a novel candidate that can kill bacteria, neutralize LPS toxicity, and inhibit the LPS-induced downstream cascade.” was changed to “Pathogenic *Escherichia coli*, a Gram-negative bacterium, causes approximately 1.7 billion cases of diarrhea in both humans and animals worldwide each year ¹. Worryingly, multidrug-resistant (MDR) *E. coli* strains were difficult to be killed by the last-resort antibiotics, polymyxins ^{2,3}. Furthermore, pathogenic Gram-negative bacterial

infections are always accompanied by the release of lipopolysaccharide (LPS) from the cell wall ^{4,5}. LPS, also known as endotoxin, is a prominent component of the outer membranes of Gram-negative bacteria and construct a prime natural barrier that restricts the entry of antibacterial agents into bacteria ^{6,7}. Additionally, LPS serves as a crucial virulence factor known as pathogen-associated molecular pattern (PAMP) ^{8,9}, which typically stimulates immune cells to activate related signaling pathways, triggering the activation of related signaling pathways and causing an uncontrollable cytokine storm. These factors contribute to the development of severe sepsis with a mortality rate of 30~50% ¹⁰⁻¹². Commonly used antibiotics, such as ampicillin and polymyxin B, cannot effectively neutralize LPS ^{13,14} due to lack of specific binding sites and LPS modifications, and even accelerate LPS release from bacteria ^{15,16}. Therefore, there is an urgent need to design a novel candidate capable of killing MDR *E. coli*, neutralizing LPS toxicity, and inhibiting the LPS-induced downstream cascade.”

The paper remains sometimes slightly hard to read.

Response: We have carefully revised the manuscript again.

Data on the bacterial strains, particularly on *E. coli* CVCC195 (except the name) and on its resistances are missing.

Response: Details of drug-resistant strains have been added to the new revised manuscript.

“^d The strains are resistant to different antibiotics.” was changed to “^d The strains are resistant to different antibiotics. The MDR *E. coli* CVCC195 strain is resistant to doxycycline, benzocillin, penicillin, lincomycin, clindamycin, vancomycin, chloramphenicol, erythromycin, amikacin, streptomycin, gentamicin and sulfamethoxazole, respectively. The *S. aureus* ATCC43300 strain is resistant to lincomycin, amoxicillin, amikacin, ampicillin, oxacillin, erythrocin, sulfisoxazole, neomycin, azithromycin, kanamycin, gentamicin, and penicillin, respectively. The *S. aureus* CVCC546 strain is resistant to tetracycline, bacitracin, and sulfisoxazole, respectively.

Other minor points:

1) The amount of furin was not given (4 μ L ? New England Biolabs), and on page 11, line 158 the molar ratio between R7 and furin should be indicated.

Response: Thank you for your comment. Furin dosage and buffer are given in the new revised draft as follow:

Chimeric peptide R7 was treated with recombinant human furin enzyme (8 U/reaction; NEB P8077) in 50- μ L reactions supplemented with in 20 mM HEPES, 0.1% Triton X-100, 1 mM CaCl₂, 0.2 mM β -mercaptoethanol. The reaction mixtures were incubated at 25°C for 65 h.

2) When parent peptides (LBP14-RKRR and L7) were tested in parallel with R7, it is unclear whether the micromolar concentration of each of these peptides was equal to that of R7 or whether the sum of the concentrations of the two peptides was equal to that of R7. This is an important issue for the proper comparison of their activities

Response: Thank you for your comment. The micromolar concentration of LBP14-RKRR and L7 was equal to that of R7, respectively. For example, 0.7 μ M “LBP14-RKRR+L7” represents a 1: 1 mixture of 0.7 μ M LBP14-RKRR and 0.7 μ M L7. To avoid misunderstanding, we made a more detailed explanation in the revised manuscript.

Page 33, Line 563 “For “LBP14-RKRR+L7” and “LBP14+L7”, the same molar amount (equal to that of R7) of parent peptides were mixed first, and then diluted by gradient.” was added in Method section.

3) Page 7 Table 1: the charge of A7 is +8 and not +10 as incorrectly reported in the table (2 E(-1) are present)

Response: Thank you for your comment. We recalculated the net charge of A7 with ExPasy protparam tool (<https://web.expasy.org/protparam/>), and its net positive charge was +10.

4) Page 10 line 136: the sentence "the survival of L7 and R7 towards RAW 264.7 cells was...." should be rewritten

Response: As suggested, this sentence has been carefully rewritten.

“at a concentration of 128 μ M, the survival of L7 and R7 towards RAW 264.7 cells was 87.7% and 58.2%” was changed to “The RAW 264.7 cell survival of L7 and R7 at the concentration of 128 μ M was 87.7% and 58.2%”

5) Page 24 lines 417-420. Results corresponding to these comments are not included in this version of the manuscript

Response: Thank you for your comment. We took the serum samples of mice injected by different routes, including intraperitoneal, intravenous or subcutaneous, and analyzed them by HPLC. The results revealed that there was no R7 signal peak, which showed a flat baseline in chromatography. Due to the large number of samples with consistently smooth baselines, here, we provided a descriptive account instead of including them directly in the manuscript: the mice were injected with R7 (10~50 mg/kg) by different routes, including intraperitoneal, intravenous or subcutaneous ones, respectively; blood was taken from the orbits at different time (5, 30, 60, and 120 min, respectively) and detected by RP-HPLC. Unfortunately, the expected target peaks of R7, L7 or LBP14-RKRR were not detected in mouse blood at any time point (data not shown)

6) Page 31. is polymycin B (PMB) actually polymyxin B?

Response: Yes, this spelling mistake has been corrected.

7) Figure S2 (page 13) Figure S2 contains the percentages of secondary structures and not the results of stability in human serum

Response: Thank you for your comment. Figure S2 in the supplementary materials is Serum stability and release of peptides in human serum. Analysis of secondary structures is Table S1 and S2 in the supplementary materials.

Reference:

1. Machholz, E., Mulder, G., Ruiz, C., Corning, B.F., Pritchett-Corning, K.R. Manual Restraint and Common Compound Administration Routes in Mice and Rats. *J. Vis. Exp.* (67), e2771,
2. Palumbo D, Iannaccone M, Porta A, Capparelli R. Experimental antibacterial therapy with puroindolines, lactoferrin and lysozyme in *Listeria monocytogenes*-infected mice. *Microbes Infect.* 2010 Jul;12(7):538-45.
3. Brezden A, Mohamed MF, Nepal M, Harwood JS, Kuriakose J, Seleem MN, Chmielewski J. Dual Targeting of Intracellular Pathogenic Bacteria with a Cleavable Conjugate of Kanamycin and an Antibacterial Cell-Penetrating Peptide. *J Am Chem Soc.* 2016 Aug 31;138(34):10945-9.
4. Ma X, Hao Y, Mao R, Yang N, Zheng X, Li B, Wang Z, Zhang Q, Teng D, Wang J. Effects of dietary supplementation of bovine lactoferrin on growth performance, immune function and intestinal health in weaning piglets. *Biometals.* 2023 Jun;36(3):587-601.

REVIEWERS' COMMENTS:

Reviewer #4 (Remarks to the Author):

I am impressed by the additional work that the authors have performed related to my comments . This has substantially improved the manuscript and removed my main concerns about the conclusions that the authors drew from the experiments. therefore I feel this version of the manuscript is acceptable for publication.